# Charge-transfer complexation of coordination cages for enhanced photochromism and photocatalysis

Gen Li[1], Zelin Du[1], Chao Wu[2], Yawei Liu[1], Yan Xu[1], Roy Lavendomme [3,4], Shihang Liang[5], En-Qing Gao [1,6] ✉ & Dawei Zhang [1,6] ✉

Intensified host-guest electronic interplay within stable metal-organic cages (MOCs) presents great opportunities for applications in stimuli response and photocatalysis. Zr-MOCs represent a type of robust discrete hosts for such a design, but their host-guest chemistry in solution is hampered by the limited solubility. Here, by using pyridinium-derived cationic ligands with tetrakis(3,5-bis(trifluoromethyl)phenyl)borate (BAr$_F^-$) as solubilizing counteranions, we report the preparation of soluble Zr-MOCs of different shapes (**1**-**4**) that are otherwise inaccessible through a conventional method. Enforced arrangement of the multiple electron-deficient pyridinium groups into one cage (**1**) leads to magnified positive electrostatic field and electron-accepting strength in favor of hosting electron-donating anions, including halides and tetraarylborates. The strong charge-transfer (CT) interactions activate guest-to-host photo-induced electron transfer (PET), leading to pronounced and regulable photochromisms. Both ground-state and radical structures of host and host-guest complexes have been unambiguously characterized by X-ray crystallography. The CT-enhanced PET also enables the use of **1** as an efficient photocatalyst for aerobic oxidation of tetraarylborates into biaryls and phenols. This work presents the solution assembly of soluble Zr-MOCs from cationic ligands with the assistance of solubilizing anions and highlights the great potential of harnessing host-guest CT for boosting PET-based functions and applications.

Metal-organic cages (MOCs)[1–3], a class of discrete metallosupramolecular capsules, are assembled from organic ligands with either metal ions or metal clusters. Their well-defined cavities enable them to have wide applications ranging from molecular recognition[4], separation[5–9], stabilization of reactive species[10], and catalysis[11–19]. Although MOCs can be used in solid state, akin to the use of metal-organic frameworks (MOFs)[20–23], it is the most appealing to make use of the molecular

attribute of MOCs that distingushes them from MOFs. In particular, MOCs can behave or be processed as discrete hollow molecules in solution, which can afford unique host-guest chemistry and functions inaccessible by porous solids.

Photoinduced electron transfer (PET) plays a central role in biological photosynthesis and many artificial photoresponsive and photocatalytic processes[24–27]. How to achieve high-efficiency PET in

[1]State Key Laboratory of Petroleum Molecular & Process Engineering, Shanghai Key Laboratory of Green Chemistry and Chemical Processes, School of Chemistry and Molecular Engineering, East China Normal University, Shanghai, China. [2]Department of Computer Science, Durham University, Durham, UK. [3]Laboratoire de Chimie Organique, Université libre de Bruxelles (ULB), Brussels, Belgium. [4]Laboratoire de Résonance Magnétique Nucléaire Haute Résolution, Université libre de Bruxelles (ULB), Brussels, Belgium. [5]State Key Laboratory of Petroleum Molecular & Process Engineering, SINOPEC Research Institute of Petroleum Processing, Beijing, PR China. [6]Institute of Eco-Chongming, Shanghai, PR China. ✉e-mail: eqgao@chem.ecnu.edu.cn; dwzhang@chem.ecnu.edu.cn

artificial systems is a great challenge. Formation of ground-state charge-transfer (CT) complexes prior to PET, in particular with the use of cage hosts to intensify host-guest electronic communications, is expected to be an effective strategy, which is explored in this work. CT complexation is capable of increasing the host-guest affinity and pre-arranging the substrate in short contact with the host, overcoming the diffusion limitation on PET[28–31]. In particular, CT complexation gives rise to bathochromic and intense photoabsorption. Excitation through the CT absorption not only increases the range and efficiency of light harvesting but also provides a more direct and faster route for PET than excitation through cage absorption.

The pyridinium unit is known for the capability of PET as well as CT due to its electron-deficient attribute[32]. It has been used as the building unit of various artificial molecules, supermolecules, polymers, and MOFs to impart photoresponsive and photocatalytic properties[33–42]. Gathering of multiple cationic pyridinium units in one coordination cage is envisioned to generate a superimposed positive field inside and around the cage and also cooperatively magnify the electron-accepting strength. The resultant hosts are thus promising to display charge-enhanced CT complexation towards electron-rich guests, activating the guests for efficient PET and leading to superior light-responsive and catalytic performances. It is thus highly desirable to prepare multipyridiniums-integrated MOCs with high solubility and stability, which remains rare.

The Zr-MOCs with $Cp_3Zr_3O(OH)_3$ vertices (Cp = $\eta^5$-$C_5H_5$) and multicarboxylic linkers, which exhibit high chemical stability due to the high Zr−O bond energy (766 kJ/mol), have witnessed rapid development since the first report in 2013 by the Yuan group[43,44]. They are usually synthesized through a conventional solvothermal approach to yield crystalline powders. The majority of Zr-MOCs have limited solubility resulting from the strong interactions between cationic $Cp_3Zr_3O(OH)_3$ vertices and counterions (generally Cl⁻), and are primarily used as crystalline solids for multiphase applications, such as gas separation[44–46] iodine capture[47], and heterogeneous catalysis[48,49]. Nevertheless, a few Zr-MOCs have been investigated as hosts for anion binding in solution or processed in solution to prepare composite membranes[50–55]. Improvement of the solubility of Zr-MOCs can be achieved by functionalizing the linkers with amino, alkyl, or other solubilizing groups[50–52]. Another rare but innovative method involves the decoration of the $Cp_3Zr_3O(OH)_3$ nodes by introducing pendant groups, such as n-butyl, benzyl, or trifluoromethylbenzyl, into the Cp rings[53,54]. A third solubilization strategy is postassembly ion exchange, which has been effective for various ionic MOCs including Zr-MOCs[55,56].

In this work, we highlight the use of counteranions to facilitate synthesis of highly soluble Zr-MOCs from pyridinium-derived cationic ligands. Different from the previous postassembly anion-exchange method[55], we introduce the solubilizing counteranion, tetrakis(3,5-bis(trifluoromethyl)phenyl)borate (BAr$_F^-$), into the flexible pyridinium-based ligands to minimize cation-anion interactions during the self-assembly, thereby preventing the formation of insoluble intermediates. Following this approach, four Zr-MOCs ranging from helicates (1-3) to tetrahedron (4) were constructed homogeneously in solutions. The incorporation of multiple electron-deficient pyridinium groups imparts superior electron-accepting abilities to the MOCs, which thus are capable of hosting electron-donating anions, including tetraarylborates, through dominative CT interactions. The ground-state interactions facilitate guest-to-host PET, resulting in efficient and regulable photochromism. In particular, the photogenerated radical state of 1 can rapidly transfer electrons to $O_2$ to generate $O_2^{·-}$, enabling the use of 1 as an effective photocatalyst for oxidation of tetraarylborate guests.

## Results

### Design, synthesis, and characterization
We first attempted to prepare the desired Zr-MOC using the conventional synthetic method for Zr cages[44], i.e., reacting zirconocene

dichloride ($Cp_2ZrCl_2$) with 1,1'-bis(4-carboxybenzyl)-4,4'-bipyridinium dichloride ($L^1$-Cl, Supplementary Fig. 1). In spite of extensive efforts to screen synthetic conditions, including solvent mixtures, temperature, pH and reaction time, this method led to insoluble amorphous powder. Considering the strong Coulombic and hydrogen-bonding interactions afforded by the chloride ions and the high flexibility of the ligand due to the presence of methylene joints, we infer that the counteranion (Cl⁻) in high content influences the self-assembly process by irreversibly forming insoluble precipitates with the cationic intermediates of the metal-organic assembly.

In order to improve the reversibility and self-correction ability during the self-assembly, BAr$_F^-$ was chosen as the counteranions of the ionic ligand. The rather bulky and highly lipophilic anion is supposed to minimize cation-anion interactions and thus to enable the high solubilities of the assembly intermediates and the product. As shown in Fig. 1a, 1-BAr$_F$ ($\{[Cp_3Zr_3(\mu_3\text{-}O)(\mu_2\text{-}OH)_3]_2(L^1)_3\}(BAr_F)_8$) was assembled from a 1:2 ratio of $L^1$·BAr$_F$ and $Cp_2ZrCl_2$ in $CH_3OH/H_2O$ at 65 °C for 12 h. The reaction gave a homogeneous solution and 1-BAr$_F$ was precipitated by adding a large amount of water. The identity of the cage was confirmed by high-resolution electrospray-ionization mass spectrometry (HR-ESI-MS), which is consistent with a $C_2L_3$ [C = cluster $Cp_3Zr_3(\mu_3\text{-}O)(\mu_2\text{-}OH)_3$, and L = Ligand] composition (Supplementary Fig. 22). The $^1H$ NMR spectrum of 1-BAr$_F$ presents only one set of ligand resonances, indicating the $C_3$-symmetry of the cage (Fig. 1b). All proton signals of 1 were assigned by two-dimensional (2D) NMR experiments (Supplementary Figs. 20–21).

1-BAr$_F$ is highly soluble in various solvents, such as methanol, acetonitrile, acetone, and DMSO (Supplementary Fig. 17). The counteranions can be easily exchanged to other anions to obtain 1-X complexes ($\{[Cp_3Zr_3(\mu_3\text{-}O)(\mu_2\text{-}OH)_3]_2(L^1)_3\}X_8$) with X = Tf$_2$N⁻ (bis(trifluoromethanesulfonyl)imide), TfO⁻ (trifluoromethanesulfonate), PF$_6$⁻, Cl⁻, or NO$_3$⁻ (Supplementary Figs. 23–24). In particular, 1-NO$_3$ is stable and highly soluble in water (Supplementary Fig. 23).

Vapor diffusion of $Et_2O$ into a MeOH solution of 1-BAr$_F$ in the presence of SCN⁻ produced crystals of 1·SCN suitable for single-crystal X-ray analysis. As shown in Fig. 2a, MOC 1 is a cage-like triple helicate with two trinuclear $[Cp_3Zr_3(\mu_3\text{-}O)(\mu_2\text{-}OH)_3]$ clusters connected by three viologen-based carboxylate linkers. The helicity arises from the gauche conformation of the linker, which is allowed by the two flexible methylene joints between viologen and benzoate moieties.

To prove the reliability of soluble Zr-MOC synthesis using solubilizing ionic ligands, we synthesized three additional Zr-MOCs of different shapes and sizes following similar procedures (Figs. 1c, 2–4). Cage 2 is isomeric to 1, with the difference being in the carboxylate position in the ligands (Supplementary Figs. 25–32). 2-BAr$_F$ ($\{[Cp_3Zr_3(\mu_3\text{-}O)(\mu_2\text{-}OH)_3]_2(L^2)_3\}(BAr_F)_8$) was found to partially dissociate into ligands and Zr-clusters in $CD_3OD$, DMSO-$d_6$, and $CD_3COCD_3$, while negligible dissociation occurred in $CD_3CN$ (Supplementary Fig. 32). The presence of additional anions, such as I⁻ and TfO⁻, could also serve as templates to drive the conversion from the assembly components into the cage (Supplementary Figs. 33–34). The assembly of a thiazolo[5,4-d]thiazole-extended viologen ligand with dibutylzirconocene dichloride (($n$-BuCp)$_2$ZrCl$_2$) led to an elongated $C_2L_3$ helicate (3-BAr$_F$, $\{[(n\text{-}BuCp)_3Zr_3(\mu_3\text{-}O)(\mu_2\text{-}OH)_3]_2(L^3)_3\}(BAr_F)_8$) (Supplementary Figs. 35–41). We noted that the assembly using $Cp_2ZrCl_2$ showed significant dissociation in $CD_3OD$, while the use of ($n$-BuCp)$_2$ZrCl$_2$ enabled the formation of 3 with negligible dissociation. This could be because the $n$-butyl groups decorating $Cp_3Zr_3O(OH)_3$ vertices further increase the solubility and stability of the cage in organic solvents[53]. A face-capped $C_4L_4$ tetrahedral cage 4-BAr$_F$ ($\{[Cp_3Zr_3(\mu_3\text{-}O)(\mu_2\text{-}OH)_3]_4(L^4)_4\}(BAr_F)_{16}$) was also successfully assembled from a tripyridinium-tricarboxylate ligand and $Cp_2ZrCl_2$ (Supplementary Figs. 42–48). These cages (2-4) have been fully characterized by 1D and 2D NMR and HR-ESI-MS. Crystals of 3-SCN were obtained by vapor diffusion of chloroform into an ethanol

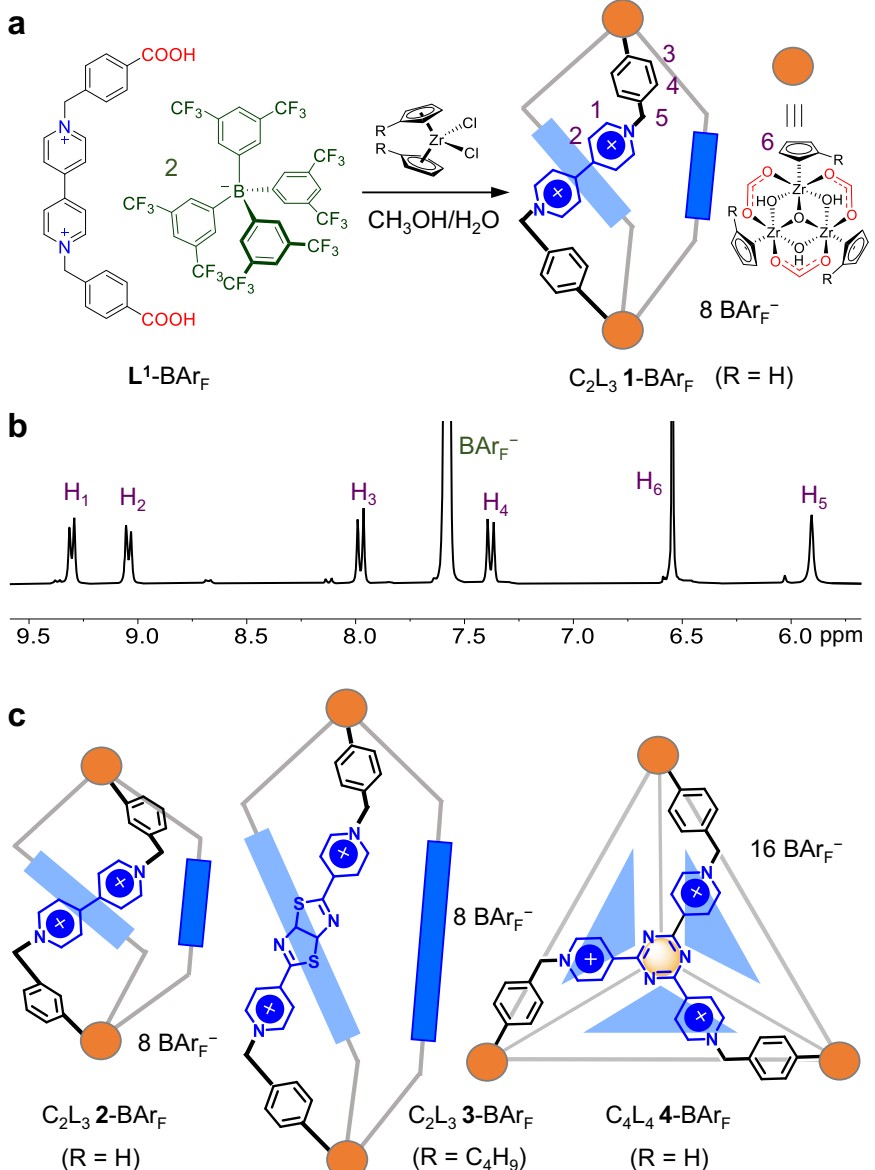

**Fig. 1 | Self-assembly of metal-organic cages. a** Self-assembly of **1**-BArF (C2L3, {[Cp3Zr3(μ3-O)(μ2-OH)3]2(L1)3}(BArF)8). **b** $^1$H NMR spectrum (CD3OD, 400 MHz, 298 K) of **1**-BArF. **c** Structures of **2**-BArF (C2L3, {[Cp3Zr3(μ3-O)(μ2-OH)3]2(L2)3} (BArF)8), **3**-BArF (C2L3, {[(n-BuCp)3Zr3(μ3-O)(μ2-OH)3]2(L3)3}(BArF)8), and **4**-BArF (C4L4, {[Cp3Zr3(μ3-O)(μ2-OH)3]4(L4)4}(BArF)16).

solution of **3**-BArF in the presence of SCN⁻. As shown in Fig. 2b, the diameter of helicate **3** is similar to that of **1**, which are determined by three methylene groups. The heights are determined by the lengths of the bipyridinium units, and the distances between the two μ3-O atoms are 24.74 Å for **3** and 19.65 Å for **1**.

## Charge transfer-promoted anion binding

Considering the very weak binding ability of BArF⁻ with the cages resulting from its bulky size, low charge density, and substituted trifluoromethyl groups[57,58], we used **1**-BArF as the host to investigate the guest binding properties of **1**. $^1$H NMR titrations of **1**-BArF with various anions (Cl⁻, Br⁻, I⁻, SCN⁻, TfO⁻, ReO4⁻, NO3⁻, ClO4⁻, PF6⁻, and Tf2N⁻, as tetrabutylammonium [TBA] salts) presented gradual shifts of resonance signals, in particular for viologen protons (H1 and H2 in Fig. 1a), indicating anion binding in fast exchange on the NMR timescale. Interestingly, the binding of the halide anions showed upfield shifts of H1 and downfield shifts of H2 (Supplementary Figs. 53–55), while other anions induced upfield shifts of both protons (Supplementary

Figs. 56–62). Binding constants were determined using BindFit (Supplementary Table 1) http://supramolecular.org/. The 1:1 binding stoichiometry was obtained for all these anions with the following binding hierarchy: I⁻ > Br⁻ > Cl⁻ > SCN⁻, TfO⁻, ReO4⁻ > NO3⁻, ClO4⁻ > PF6⁻, Tf2N⁻. The binding sequence for halides is not determined by Coulombic interactions or charge localization of halides but consistent with their electron-donor (ED) strength, suggesting the dominance of electron donor-acceptor CT interactions between halides and the viologen moieties of the cage. The strong CT binding of I⁻ and Br⁻ is evidenced by the visual color changes (from colorless to yellow) and the appearance of CT absorption in UV-Vis spectra upon addition of these anions into **1**-BArF (Supplementary Fig. 68). Weak CT interactions between **1** and Cl⁻ or SCN⁻ were confirmed by UV-Vis spectra. The rest of the anions show no indication of CT due to their poor ED abilities.

Crystal structures of the host-guest complexes revealed the exact positioning of the bound anions within **1**. Single crystals of **1**·I and **1**·Br were obtained by slow vapor diffusion of chloroform or Et2O into EtOH or MeOH solutions of **1**-BArF in the presence of TBAI or TBABr. For **1**·I

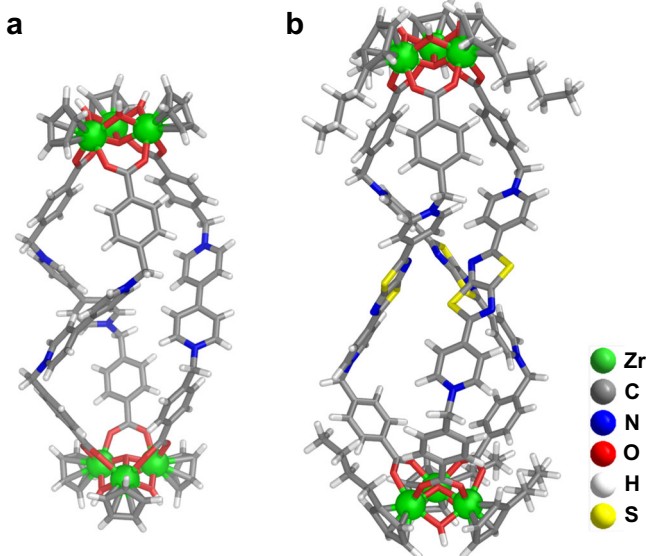

**Fig. 2 | X-ray crystallographic structures of cages.** Structures of **1** in **1**-SCN (**a**) and **3** in **3**-SCN (**b**).

(Fig. 3a), an I⁻ anion resides at the center of the helicate host, maintaining a pseudo-$C_3$ symmetry. The host-guest interactions involve a set of six C-H•••I⁻ (2.50 – 3.32 Å for H•••I) contacts between the I⁻ and three viologen units. For **1**-Br (Fig. 3b), a Br⁻ anion is also encapsulated inside the host but significantly displaced from the center, interacting with the host through three C-H•••Br⁻ (2.66 – 2.75 Å for H•••Br) bonds with two viologen units and an anion-π contact (3.26 Å) with the third viologen unit. The offset guest positioning inside the cage leads to an asymmetric gourd-shaped structure, with a linker swinging inwards for the anion-π interaction. The different host-guest structures of **1**-I and **1**-Br reflect the guest adaptability of the cage arising from the flexible linkers. In both structures, multiple pyridinium units from the cage cooperate in interacting with the guest ions, which accounts for the large binding constants and the strong CT complexation.

Tetraarylborate anions with variable ED abilities also provide an opportunity to investigate the CT-based binding properties of the electron-poor cage. Five tetraarylborates, [B(Ph-R)₄]⁻ with R = OCH₃, CH₃, H, Cl, and F (in an order of descending ED strength, Fig. 4b) were chosen. Significant signal shifts in ¹H NMR were observed upon adding tetraarylborates into **1**-BArF in CD₃OD. For instance, addition of BPh₄⁻ resulted in notable upfield shifts for H₁, H₂, H₄, and H₅ of **1** (Fig. 4a), indicating guest-induced conformational and/or charge-density changes of **1**. The peaks of BPh₄⁻ also experience a significant upfield

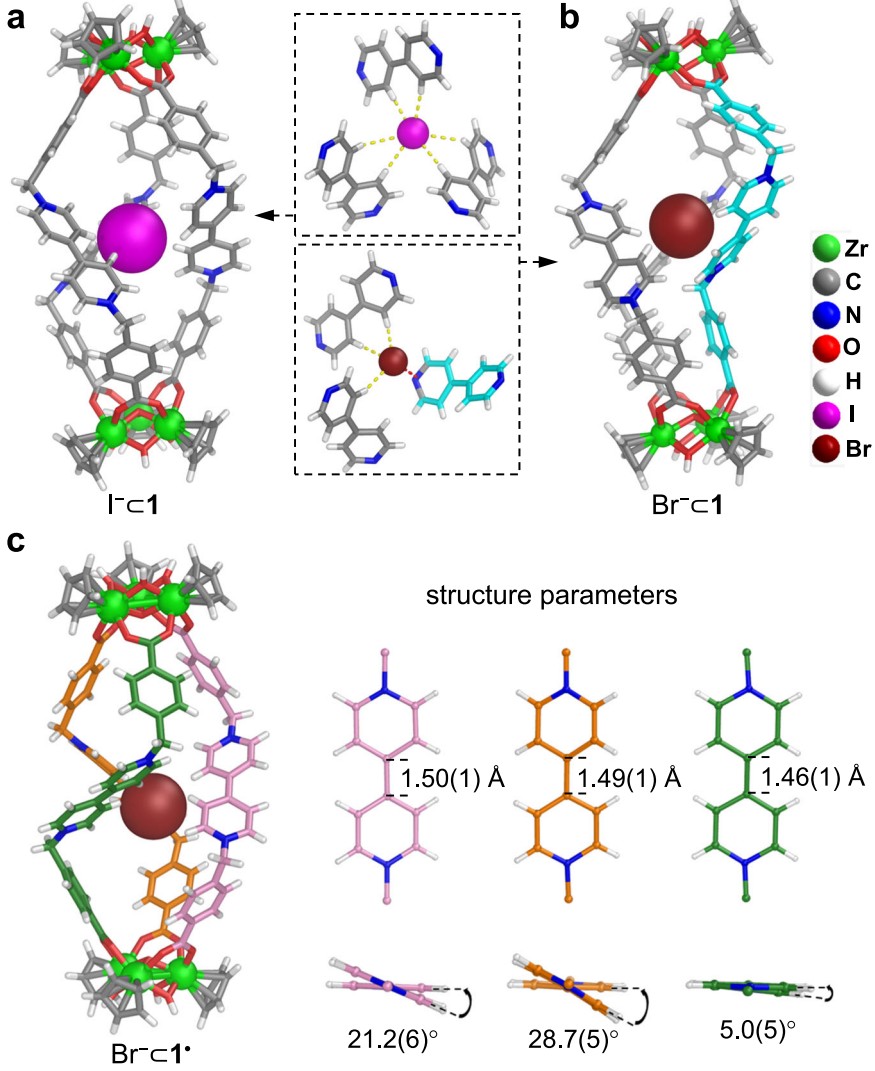

**Fig. 3 | X-ray crystallographic structures of host-guest complexes. a** Structure of I⁻⊂**1** in **1**-I. **b** Structure of Br⁻⊂**1** in **1**-Br. The imploded viologen ligand is colored cyan. **c** Structure of Br⁻⊂**1·** in **1·**-Br and the structure parameters of the three viologen units showing the radical state of the cage.

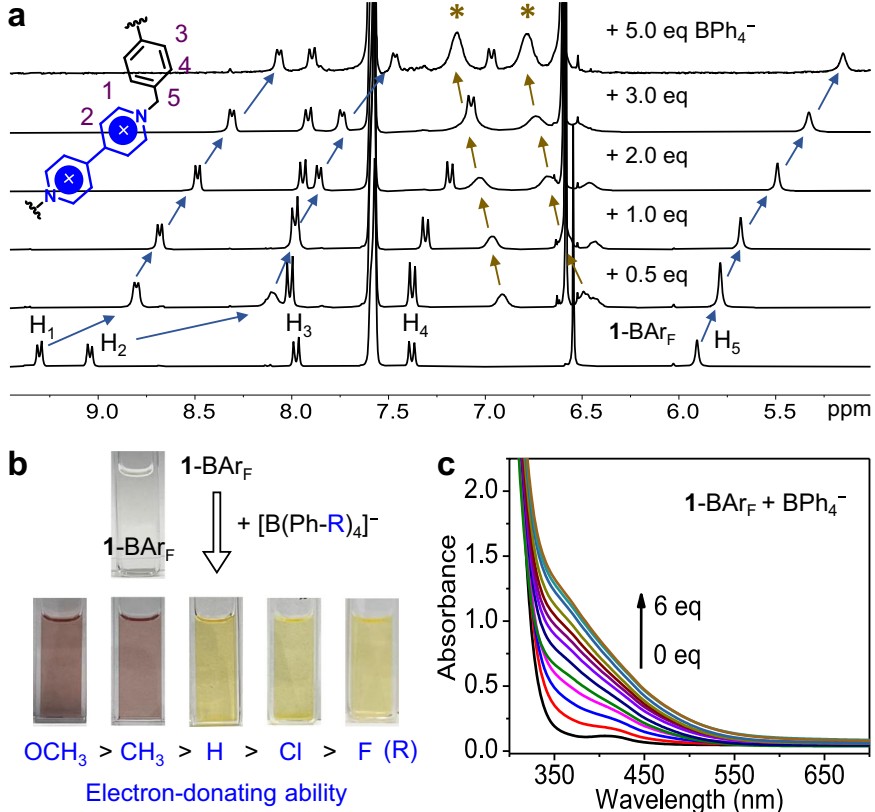

**Fig. 4 | Guest binding studies of 1 towards tetraarylborates. a** $^1$H NMR (400 MHz, CD$_3$OD, 298 K) titrations of BPh$_4^-$ into **1**-BAr$_F$. Peaks from the guest are indicated by asterisks. **b** Color of the methanol solution of **1**-BAr$_F$ in the absence or presence of 6 equiv. [B(Ph-R)$_4$]$^-$. **c** UV-vis spectrophotometric titrations of BPh$_4^-$ into a methanol solution of **1**-BAr$_F$ (0.1 mM).

shift upon adding 0.5 equiv. compared to the signals of free BPh$_4^-$, and present gradual downfield shifts with increasing its concentration, consistent with the shielding effects of the anisotropic cones of the polyaromatic cage on the bound guest. Similar phenomena were also observed for other tetraarylborate anions (Supplementary Figs. 63–67) and there is an overall trend that the anions with higher ED strength cause larger shifts for the protons of **1**. The binding constants obtained by $^1$H NMR titrations (Supplementary Table 2) are also positively correlated with the ED strength of tetraarylborates, supporting the predominance of CT interactions.

The CT complexation is evidenced by the distinctive chronic response of **1**-BAr$_F$ towards tetraarylborates (Fig. 4b). The colorless solution of **1**-BAr$_F$ turned yellow upon addition of BPh$_4^-$, [B(Ph-Cl)$_4$]$^-$, or [B(Ph-F)$_4$]$^-$, whereas the addition of [B(Ph-OCH$_3$)$_4$]$^-$ or [B(Ph-CH$_3$)$_4$]$^-$ gave rise to maroon solutions. UV–vis spectrophotometric titrations of **1**-BAr$_F$ with the former three tetraarylborates present the gradual emergence of a broad absorption extending into the visible-light region (Fig. 4c and Supplementary Fig. 69c–d), which is assignable to guest-to-host CT transitions. For [B(Ph-OCH$_3$)$_4$]$^-$ and [B(Ph-CH$_3$)$_4$]$^-$, the CT absorption bands extend longer into the visible-light region (Supplementary Fig. 69a–b).

### Photochromism and generation of superoxide

Upon exposure to Xe lamp, the solution of **1**-BAr$_F$ in air-free methanol changed from colorless to blue (Fig. 5a). The UV-vis spectra showed new bands centered around 400 and 612 nm, which are characteristic of the viologen radical (V$^{·+}$)[59]. The bands grew with irradiation time and reached saturation after 180 s. The photochromic response involves a PET process, in which the viologen unit V$^{2+}$ accepts an electron to generate the blue-colored radical V$^{·+}$ (Fig. 5c). The radical formation was further confirmed by the strong ESR (electron spin resonance) signal at

$g = 2.0040$ (Fig. 5d). The photochromic phenomenon was not observed for **1**-OTf but was for the mixture of **1**-OTf and NaBAr$_F$ (Supplementary Fig. 70), which evinced that BAr$_F^-$ serves as the ED in PET.

Considering the incapability of TfO$^-$ as an ED, the PET behaviors of different tetraarylborates with **1** can thus be conveniently compared by adding [B(Ph-OCH$_3$)$_4$]$^-$, BPh$_4^-$, or BAr$_F^-$ into solutions of **1**-OTf. Results showed that the PET-based photochromic performance of the mixtures is positively correlated to the ED strengths and CT capability of tetraarylborates (Fig. 5b and Supplementary Fig. 71). BAr$_F^-$ gave the slowest and weakest response, and the response with [B(Ph-OCH$_3$)$_4$]$^-$ was the fastest and the strongest.

The photogenerated blue solutions (Fig. 5a, b), either from **1**-BAr$_F$ or from mixed **1**-OTf and BAr$_4^-$, showed no indication of fading if kept under N$_2$ overnight, but upon exposure to O$_2$ or air, the solutions faded rapidly concomitant with the generation of biaryls and phenols (oxidation products of BAr$_4^-$) (Supplementary Fig. 72). The phenomenon indicates that the blue V$^{·+}$ radical was quenched through electron transfer (ET) to O$_2$, which generated superoxide (O$_2^{·-}$) for BAr$_4^-$ oxidization (*vide infra*). The generation of O$_2^{·-}$ was confirmed by spin-trapping ESR for a sample of **1**-BAr$_F$ irradiated under O$_2$. The use of 5,5-dimethyl-1-pyrrolin N-oxide (DMPO) as spin trap led to the sharp ESR signals of the DMPO·O$_2^{·-}$ adduct (Fig. 5e).

The successive PET and ET processes of **1**-BAr$_F$ were also studied in other solvents. According to the UV-Vis spectra at different irradiation time, the photochromic contrast and kinetics of **1**-BAr$_F$ in different solvents (deoxygenated) present the following order: CH$_3$COCH$_3$ > DMSO > CH$_3$CN ~ CH$_3$OH (Supplementary Figs. 73-74). This indicates the highest PET efficiency in acetone. Interestingly, the fading rates of the blue radical solutions in air follow an approximately inverse order: CH$_3$COCH$_3$ < DMSO < CH$_3$CN < < CH$_3$OH (Fig. 5f), based on the ratio of $A_t/A_0$, where $A_0$ and $A_t$ represent absorbances of the

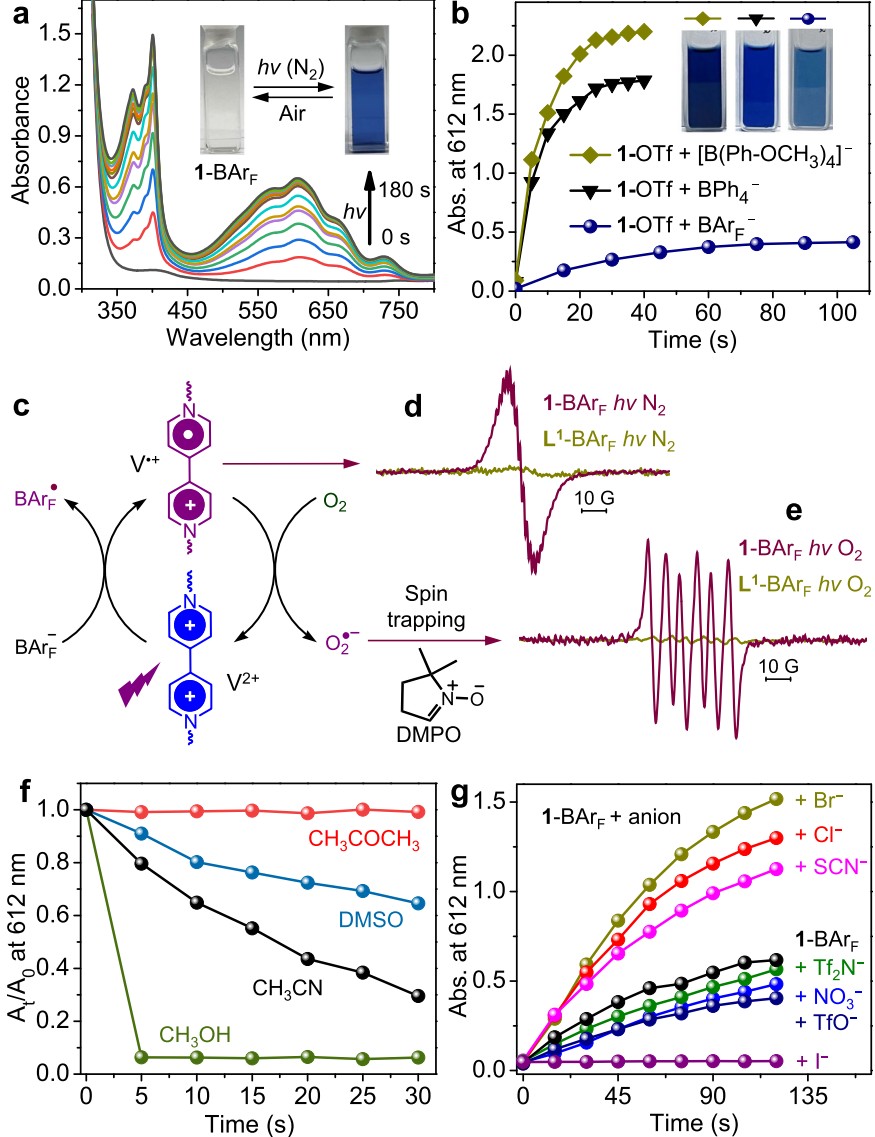

**Fig. 5 | Photochromism studies of 1 towards tetraarylborates. a** UV-vis spectra of **1**-BAr$_F$ (190 μM) in air-free methanol upon Xe lamp irradiation for different periods of time. Insets: the reversible color changes upon irradiation and exposure to air. **b** Photochromic kinetics of **1**-OTf (65 μM) in air-free methanol in the presence of 6 equiv. tetraarylborates. Insets: saturation color of the solutions. **c** Scheme illustrating PET from BAr$_F^-$ to V$^{2+}$ and ET from V$^{+}$ to O$_2$. **d** ESR spectra (CH$_3$OH, 9.82 GHz, 298 K) of **1**-BAr$_F$ and **L**$^1$-BAr$_F$ irradiated under oxygen-free conditions. **e** ESR spectra (CH$_3$OH, 9.82 GHz, 298 K) of **1**-BAr$_F$ and **L**$^1$-BAr$_F$ irradiated in the presence of O$_2$ and DMPO. **f** Solvent effect on color-fading kinetics of irradiated **1**-BAr$_F$ (190 μM) upon exposure to air. **g** Anion effect (8 equiv.) on photochromism of **1**-BAr$_F$ (190 μM) in CH$_3$OH.

irradiated **1**-BAr$_F$ before and after exposure to air. In particular, the acetone solution did not fade overnight in air, while the methanol solution faded completely within 5 s. The fast fading in methanol indicates a high ET activity of V$^{+}$ in the solvent and a high superoxide-generation efficiency, which is conducive to photocatalytic oxidation.

Apart from solvent, the PET behaviors of **1**-BAr$_F$ can also be modulated by coexistent anions, showing a weakening, enhancement, or suppression effect (Fig. 5g). Upon introducing anions with poor ED ability, such as TfO$^-$, NO$_3^-$, or Tf$_2$N$^-$, the photochromic response of **1**-BAr$_F$ was depressed, as indicated by the slowed growth of the characteristic radical absorptions with irradiation time (Supplementary Fig. 75). The weakening effect is attributed to the binding of these anions to the cage, which expels BAr$_F^-$ away from the cage to disadvantage the PET from BAr$_F^-$.

In contrast, the addition of SCN$^-$, Cl$^-$, or Br$^-$ to **1**-BAr$_F$ significantly enhances the photochromic response (Supplementary Fig. 76). The degree of enhancement increases in the order SCN$^-$ < Cl$^-$ < Br$^-$ (Fig. 5g),

which is consistent with the increase of their binding constants as well as CT capabilities with **1**. The enhancement effect can be attributed to the dominance of these electron-rich (pseudo)halide ions in PET as EDs. Differently, although I$^-$ has the highest ED strength and also the largest binding constant with **1**, the photochromic response of **1**-BAr$_F$ was significantly diminished by I$^-$ and even completely suppressed when in the presence of more than 4 equiv. of I$^-$. (Supplementary Fig. 77). The strongly bound I$^-$ anions can prevent BAr$_F^-$ from accessing the cage, and the strong ground-state CT interaction between I$^-$ and V$^{2+}$ can impair the electron deficiency of V$^{2+}$, suppressing PET from BAr$_F^-$. Moreover, the firm CT complexation allows not only ultrafast PET from I$^-$ to V$^{2+}$ but also ultrafast back ET in the picosecond time scale, precluding a sufficiently long lifetime of the photogenerated V$^{+}$ to display its color[39,59].

X-ray crystal structures of the cage after photochromism were determined. Slow evaporation of Et$_2$O into a fully irradiated methanol solution of **1**-BAr$_F$ in the presence of Br$^-$ or SCN$^-$ under nitrogen led to

**Table 1 | Photocatalytic transformation of BPh$_4^-$**

| Entry | Catalyst | Solvent | Atmos. | Conv. (%)[a] |
|---|---|---|---|---|
| 1 | **1**-NTf$_2$ | CH$_3$OH | O$_2$ | 92 (100[b]) |
| 2 | **1**-NTf$_2$ | CH$_3$OH | air | 81 |
| 3 | / | CH$_3$OH | O$_2$ | trace |
| 4[c] | **1**-NTf$_2$ | CH$_3$OH | O$_2$ | trace |
| 5 | **1**-NTf$_2$ | CH$_3$OH | N$_2$ | 6 |
| 6[d] | Cp$_2$ZrCl$_2$ | CH$_3$OH | O$_2$ | trace |
| 7[e] | **L$^1$**-NTf$_2$ | CH$_3$OH | O$_2$ | 31 |
| 8 | **1**-NTf$_2$ | CH$_3$COCH$_3$ | O$_2$ | 43 |
| 9 | **1**-NTf$_2$ | DMSO | O$_2$ | 51 |
| 10 | **1**-NTf$_2$ | CH$_3$CN | O$_2$ | 76 |
| 11 | **1**-OTf | CH$_3$OH | O$_2$ | 87 |
| 12 | **1**-Cl | CH$_3$OH | O$_2$ | 78 |
| 13[f] | **1**-BPh$_4$ | CH$_3$OH | O$_2$ | 98 |
| 14 | **1**-NO$_3$ | H$_2$O | O$_2$ | 88[b] (99[g]) |

[a]The conversion is equal to the yields of biphenyl and phenol (1:2 molar ratio) determined by $^1$H NMR.
[b]Reaction time 12 h.
[c]No light.
[d]Catalyst amount: 60 mol%.
[e]Catalyst amount: 30 mol%.
[f]**1**-BPh$_4$ (8 BPh$_4^-$ ions per cage) and NaBPh$_4$ were used in 1:2 molar ratio so that the amount of the cage catalyst was 10 mol%. The yield was calculated based on the total amount of BPh$_4^-$.
[g]Reaction time 18 h.

single crystals of **1'**-Br and **1'**-SCN. The crystals show a dark blue color characteristic of V$^{\cdot+}$. While eight anions are required for charge balance of a nonradical cage, ion chromatography revealed the existence of seven and six counteranions per cage for **1'**-Br and **1'**-SCN, respectively. The anion contents indicate that 1/3 and 2/3 of the viologen units in the two compounds are V$^{\cdot+}$. Crystallographic analysis allowed the discrimination of V$^{\cdot+}$ from V$^{2+}$ in **1'**-Br. As shown in Fig. 3c, the interannular torsion angles and the interannular C-C bond distances of two viologen units in **1'**-Br are similar to those of V$^{2+}$ in **1**-Br (25.3–40.6° and 1.47–1.50 Å, respectively). However, the third viologen unit in **1'**-Br shows a much smaller torsion angle and a shorter C-C bond (5.0° and 1.46 Å, Fig. 3c), which can be ascribed to V$^{\cdot+}$. The differences reflect the increased planarity and bond order between the two pyridinium rings from V$^{2+}$ to V$^{\cdot+}$[60]. Compared with **1**-Br, **1'**-Br also presents significant changes in cage shape and host-guest interactions. Different from the gourd-like shape and the off-center Br$^-$ encapsulation of the cage in **1**-Br (Fig. 3b), the cage in **1'**-Br adopts the quasiregular helical shape and encapsulates Br$^-$ at the center through hydrogen bonding interactions (Supplementary Fig. 51).

The overall structure of the cage in **1'**-SCN (Supplementary Fig. 52a) is similar to that in **1**-SCN (Fig. 2a), exhibiting the helical shape with no anions inside. Differently, V$^{\cdot+}$ and V$^{2+}$ in **1'**-SCN cannot be crystallographically differentiated. Nevertheless, in the radical-containing cage, the average inter-annular torsion angle is decreased and the average inter-annular C-C bond is shortened (Supplementary Fig. 52b).

## Photocatalytic oxidation of tetraarylborates
The efficient PET and superoxide-generation capabilities of **1** with tetraarylborates provide an opportunity for photocatalytic oxidation

of tetraarylborates. Chemical, electrochemical, and photochemical oxidation of tetraarylborates has been studied as new approaches towards biaryls[61–67]. In these cases, biaryls are usually generated by coupling two of the four aryl rings in the substrates, with the byproduct having been reported to be diarylborinic acids or left unidentified. The use of **1** as a photocatalyst enables simultaneous oxidative coupling and hydroxylation, generating biaryls and phenols in the 1:2 molar ratio, with no other side products. Note that both products are useful building blocks in chemical and pharmaceutical industries.

We first investigated the oxidation of NaBPh$_4$ in methanol with **1**-NTf$_2$ as the photocatalyst. **1**-NTf$_2$ was chosen because Tf$_2$N$^-$, as a CT- and PET-inactive anion showing the weakest binding with **1**, is not expected to perturb the interactions of **1** with substrates. Under mild conditions (room temperature, O$_2$ balloon, 6 W light at 400 nm), the use of 10 mol% **1**-NTf$_2$ gave rise to excellent conversion to biphenyl and phenol (entry 1 in Table 1). The reaction in air also gave satisfactory results (entry 2). In the absence of any catalyst, light, or dioxygen, no effective conversion was observed (entries 3-5), which confirms the catalytic role of **1**-NTf$_2$ in the light-driven aerobic oxidation reaction. The inactivity of Cp$_2$ZrCl$_2$ (entry 6) suggests the catalytic activity of **1**-NTf$_2$ arises from the organic linker rather than the metal center.

Catalytic performance of **1**-NTf$_2$ in different solvents increases in the order CH$_3$COCH$_3$ < DMSO < CH$_3$CN < CH$_3$OH (entries 1 and 8-10). The solvent effect is consistent with the solvent dependence of the color-fading kinetics observed in photochromic studies (Fig. 5f): the solvent that affords faster V$^{\cdot+}$-to-O$_2$ ET allows faster regeneration of V$^{2+}$ for the next cycle of PET with the substrate and also faster generation of O$_2^{\cdot-}$ for oxidation. Photoconversion of BPh$_4^-$ catalyzed by **1** with different anions decreases as **1**-NTf$_2$ > **1**-OTf > **1**-Cl (entries 1 and 11-12,

**Table 2 | Photocatalytic transformation of various tetraarylborates**

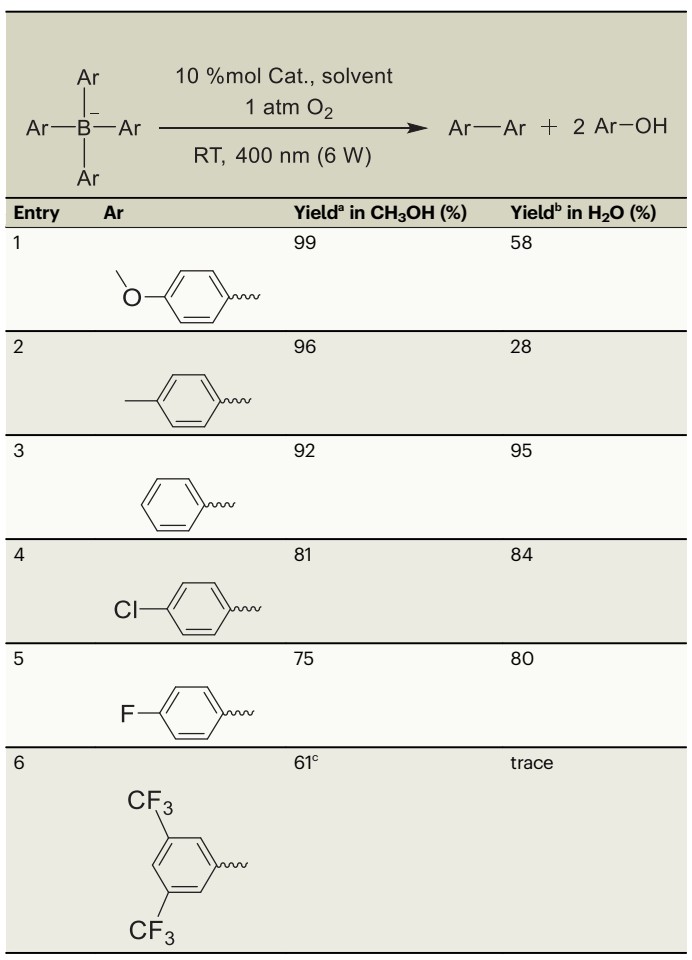

| Entry | Ar | Yield[a] in CH$_3$OH (%) | Yield[b] in H$_2$O (%) |
|---|---|---|---|
| 1 | (4-OCH$_3$-phenyl) | 99 | 58 |
| 2 | (4-CH$_3$-phenyl) | 96 | 28 |
| 3 | (phenyl) | 92 | 95 |
| 4 | (4-Cl-phenyl) | 81 | 84 |
| 5 | (4-F-phenyl) | 75 | 80 |
| 6 | (3,5-(CF$_3$)$_2$-phenyl) | 61[c] | trace |

Reaction conditions: Ar$_4$B$^-$ with 10 %mol Cat., solvent, 1 atm O$_2$, RT, 400 nm (6 W) → Ar—Ar + 2 Ar—OH

[a]**1**-NTf$_2$ as the catalyst; reaction time 10 h.
[b]**1**-NO$_3$ as the catalyst; reaction time 15 h.
[c]Reaction time 48 h. The yields of biaryls and phenols are equal in each case determined by $^1$H NMR.

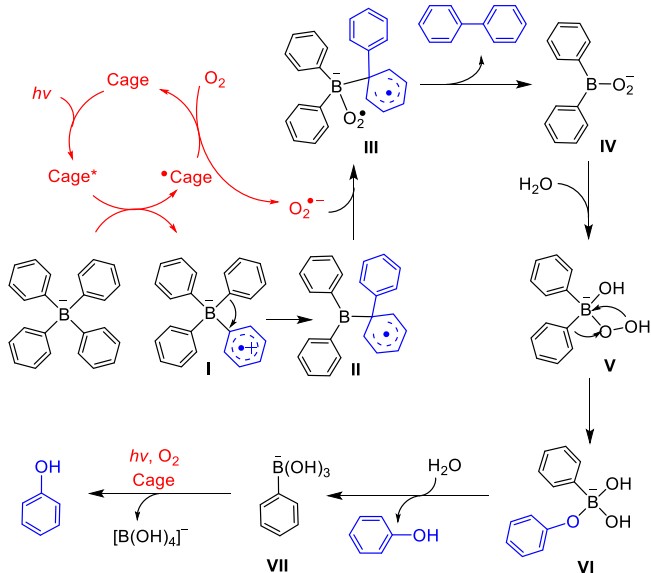

**Fig. 6 | Proposed mechanism for photocatalytic transformation of tetraphenylborate with 1. 1** undergoes PET with BPh$_4^-$ to produce BPh$_4^{\cdot}$ (**I**) and subsequent ET with O$_2$ to produce O$_2^{\cdot-}$. The intramolecular C-C coupling through 1,2-rearrangement in **I** leads to intermediate **II**, to which O$_2^{\cdot-}$ is added to give **III**. Biphenyl product is released from **III**, along with intramolecular ET to coordinated O$_2^-$ generating a peroxide intermediate **IV**. **IV** reacts with water to give **V**, which then undergoes B-to-O phenyl migration to give **VI**. Hydrolysis of **VI** produces phenol product and **VII**, the latter of which further undergoes oxidative hydroxylation in the presence of **1** to produce another equivalent of phenol.

Supplementary Fig. 78). The anion dependence can be related to the binding affinity (Supplementary Table 1): the competing anion with strong affinity adversely influences the interactions of BPh$_4^-$ with **1**, and the electron-donating anion like Cl$^-$ also competes with BPh$_4^-$ in PET. When **1**-BPh$_4$ was used as the source of the catalytic cage, the conversion was slightly higher than that using **1**-NTf$_2$ (entry 13), confirming a weak adverse effect of the Tf$_2$N$^-$ ion.

Impressively, water-soluble **1**-NO$_3$ allows the reaction to be conducted in water with excellent yields (entry 14 in Table 1). Notably, the water-insoluble biphenyl product precipitates directly from the aqueous solution, allowing facile separation. These results complied with the principles of green chemistry: molecular oxygen as oxidant, water as solvent, high selectivity, and facile isolation of products.

To verify the reliability of **1** as the photocatalyst, tetraarylborates with varying substitution groups were tested (Table 2). All substrates underwent simultaneous oxidative coupling and hydroxylation to give biaryls and phenols. For [B(Ph-R)$_4$]$^-$ with **1**-NTf$_2$ in CH$_3$OH, the conversion after 10 h varies in the following order: R = OCH$_3$ > CH$_3$ > H > Cl > F. BAr$_F^-$ shows the lowest reactivity and requires longer time. The order, also observed in their binding hierarchy with **1** and photochromic response, is in good agreement with ED strength of the substrates. The oxidation of [B(Ph-R)$_4$]$^-$ (R = H, Cl or F) with **1**-NO$_3$ in water also gave satisfactory yields (>80%, 15 h). The low conversions for [B(Ph-OCH$_3$)$_4$]$^-$ and [B(Ph-CH$_3$)$_4$]$^-$ and the trace conversion for BAr$_F^-$ result from their low solubility in water (Table 2).

To determine whether the aryl coupling is intermolecular or intramolecular, the photocatalytic reaction was performed with a mixture of [B(Ph-OCH$_3$)$_4$]$^-$ and BPh$_4^-$ (Supplementary Fig. 79). Only homocoupling biaryl products (biphenyl and 4,4'-dimethoxybiphenyl) were obtained, which supports intramolecular coupling.

Based on the investigations and previous reports for oxidation of tetraarylborates and other organoboron[65,66], the catalytic mechanism is proposed in Fig. 6. The cage undergoes PET with bound BPh$_4^-$ to produce BPh$_4^{\cdot}$ (**I**) and subsequent ET with O$_2$ to produce O$_2^{\cdot-}$. The intramolecular C-C coupling through 1,2-rearrangement in **I** leads to intermediate **II**[67], to which O$_2^{\cdot-}$ is added to give **III**. The release of biphenyl product from **III** and concomitant intramolecular ET to coordinated O$_2^{\cdot-}$ generate a peroxide intermediate (**IV**), which readily reacts with water to give **V**. **V** undergoes B-to-O phenyl migration and subsequent hydrolysis to produce phenol product and phenylboronate (**VII**)[68]. Finally, **VII** undergoes oxidative hydroxylation with **1** as the photocatalyst to produce another equivalent of phenol, which has been verified by the photocatalytic tests starting directly with NaBPh(OH)$_3$ (Supplementary Fig. 80).

## Performances of cage in comparison to ligand and CT contribution

Ligand **L$^1$**-BAr$_F$ exhibited no appreciable $^1$H NMR shifts upon introducing an excess of TfO$^-$, I$^-$, or BPh$_4^-$ (Supplementary Figs. 81–82), suggesting interactions of these anions with the free ligand are very weak. Addition of electron-donating anions, such as BPh$_4^-$, [B(Ph-OCH$_3$)$_4$]$^-$, and I$^-$ into **L$^1$**-BAr$_F$ led to only faint changes in color and UV-Vis spectra (Fig. 7a, b and Supplementary Figs. 83–84), proving weak CT complexation. The properties of **L$^1$**-BAr$_F$, contrary to those of **1**-BAr$_F$, demonstrate that the gathering of multiple viologen linkers in one cage leads to much enhanced CT binding capabilities.

In contrast to the pronounced photochromic response of **1**-BAr$_F$ or mixed **1**-NTf$_2$ and BPh$_4^-$, Xe-light irradiation of **L$^1$**-BAr$_F$ or mixed **L$^1$**-NTf$_2$ and BPh$_4^-$ led to much fainter blue colors as well as much weaker

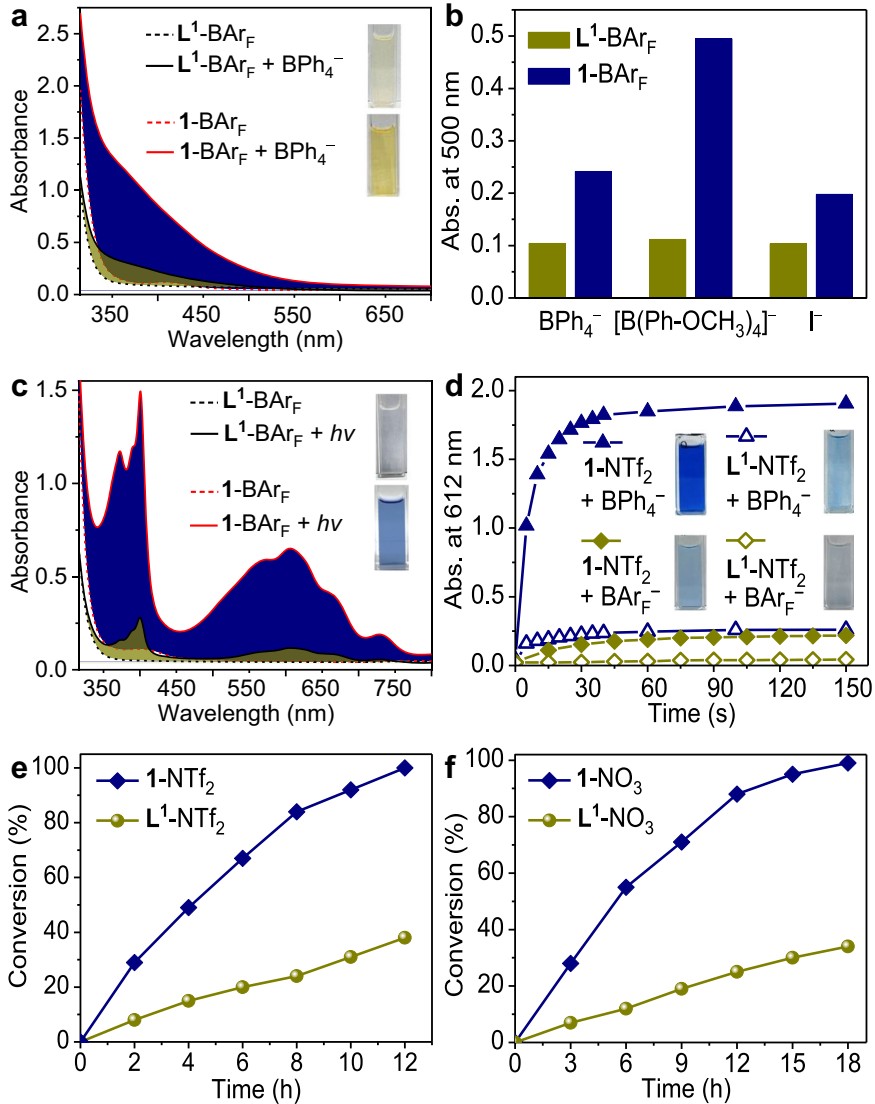

**Fig. 7 | Comparison of performances in CT complexation, photochromism, and photocatalysis between cage and ligand. a** UV-vis spectra of **1**-BAr$_F$ (100 µM) and **L¹**-BAr$_F$ (300 µM) before and after addition of BPh$_4^-$ (600 µM) in CH$_3$OH. The filled areas show the absorption increase after BPh$_4^-$ addition. Insets: color after BPh$_4^-$ addition. **b** Comparison of the absorbances for **1**-BAr$_F$ (100 µM) and **L¹**-BAr$_F$ (300 µM) in the presence of BPh$_4^-$ (600 µM), [B(Ph-OCH$_3$)$_4$]$^-$ (600 µM), or I$^-$ (800 µM) in CH$_3$OH. **c** UV-vis spectra of **1**-BAr$_F$ (190 µM) and **L¹**-BAr$_F$ (570 µM) in CH$_3$OH before and after Xe-light irradiation for 150 s. The filled areas show the absorption increase after irradiation. Insets: color after irradiation. **d** Photochromic kinetics of the mixture of **1**-NTf$_2$ (65 µM) or **L¹**-NTf$_2$ (195 µM) with BPh$_4^-$ (390 µM) or BAr$_F^-$ (390 µM) in CH$_3$OH. Insets: color after irradiation. **e, f,** Kinetics for photooxidation of BPh$_4^-$ catalyzed by **1** or **L¹** in CH$_3$OH (**e**) and in H$_2$O (**f**). Conditions: 10 mol% for **1** or 30 mol% for **L¹**, O$_2$, 6 W light at 400 nm.

radical signals in ESR and UV-Vis spectra (Figs. 5d and 7c, d). The above results prove that assembling the ligand into the cage causes remarkable enhancements in PET efficiency. To quantitatively evaluate the cage effect, PET enhancement factors (EFs) were calculated according to the ratio of the radical absorbances of the cage and the ligand after irradiation (A$_C$/A$_L$). The factor represents a measure of the concentration ratio of V$^{•+}$ generated from the cage against that from the free ligand. The factors are up to 5.1 for BAr$_F^-$ and 7.4 for BPh$_4^-$ after irradiation for 150 s (Supplementary Figs. 85–87). Since BAr$_F^-$ is a poor ED showing no effective CT complexation with either the cage or the ligand, the PET enhancement could be associated with the high positive charge of the cage, which facilitates the access and contact of BAr$_F^-$ for PET. The even larger EF observed for BPh$_4^-$ than BAr$_F^-$ highlights the significant contribution of host-guest CT complexation to the PET, which can prevent diffusion dependence of PET.

Efficient PET is a prerequisite for efficient superoxide-generation through ET from V$^{•+}$ to O$_2$. Spin-trapping ESR tests with the ligand

showed very weak DMPO·O$_2^{•-}$ signals, in contrast to the strong signals observed with the cage (Fig. 5e). The comparison evinces that the ability to produce O$_2^{•-}$ is greatly enhanced after cage formation.

To evaluate the cage effect on photocatalysis, kinetic experiments were carried out for photooxidation of BPh$_4^-$. As shown in Fig. 7e-f, the reaction with **1**-NTf$_2$ (in methanol) or **1**-NO$_3$ (in water) as catalyst proceeded much faster than the reaction catalyzed by the corresponding ligand. When the conversion of BPh$_4^-$ with the cage was completed, only one third of the substrate was converted by the ligand. In the first 2 h, the turnover frequencies (TOFs) over **1**-NTf$_2$ and **1**-NO$_3$ are higher than those over the ligands by a factor of 3.6 and 4.0, respectively. Note that oxidative coupling of BAr$_F^-$ has succeeded with electrochemical method[62] but failed with chemical oxidants[64,65] for its high oxidation potential. Our results show that BAr$_F^-$ can be oxidized by O$_2$ using **1**-NTf$_2$ as photocatalyst, although prolonged time is needed (Table 2). However, the use of **L¹**-NTf$_2$ only led to a trace conversion, demonstrating the high potency of the MOCs in upgrading photocatalytic performances.

## Discussion

In conclusion, introduction of $BAr_F^-$ as the counteranions of cationic ligands enable homogeneous assembly processes in solutions, resulting in soluble Zr-MOCs. Helicate-like cage **1** is capable of binding a variety of anions, which is enabled by the Coulombic interactions for anions with poor ED strength and dominated by CT complexation for electron-donating halides and tetraarylborates. In comparison to the free ligand, the CT complexation is significantly enhanced owing to the cooperation of the multiple cationic viologen groups in the cage. The enhanced binding leads to pronounced photochromism due to anion-to-viologen PET, which can be modulated by coexistent anions and was characterized by X-ray crystal structures. The photo-generated viologen radical readily undergoes ET to dioxygen to produce superoxide. Combining the efficient PET and superoxide-generation properties, **1** can effectively catalyze photooxidation of tetraarylborates, which allows the simultaneous synthesis of biaryls and phenols through aryl coupling and hydroxylation. This work demonstrates the great potential of using host-guest CT complexation of MOCs to manipulate PET for sophisticated responsive and catalytic performances. Extended exploration with various electron-deficient soluble Zr-MOCs (e.g., **2-4**) is underway in our laboratory.

## Methods

### Synthesis of 1-BAr_F

$L^1$-$BAr_F$ (20 mg, 9.3 μmol) and $Cp_2ZrCl_2$ (5.8 mg, 19 μmol) were added into a solution mixture containing 3.0 mL $CH_3OH$ and 0.12 mL $H_2O$. The reaction mixture was stirred and kept at 65 °C overnight. After cooling to room temperature, 3.0 mL $H_2O$ was added and a large amount of white precipitate appeared. The precipitate was collected through centrifugation, thoroughly washed with water (5 mL × 3), and dried under vacuum to obtain **1**-$BAr_F$ (21 mg, 74% yield).

### Synthesis of 1-X (X = NO_3, OTf, PF_6, Cl, and NTf_2)

**1**-$BAr_F$ (50 mg, 5.4 μmol) was dissolved in 2.0 mL $CH_3OH$. Upon addition of $TBANO_3$ (24 mg, 81 μmol), a large amount of white precipitate appeared immediately. The precipitate was collected by filtration, washed with $CH_3OH$ (3.0 × 10 mL), and dried under vacuum to obtain **1**-$NO_3$ (81% yield).

**1**-$NO_3$ (50 mg, 17 μmol) was dissolved in 5.0 mL $H_2O$. Upon addition of TBAX (X = OTf, $PF_6$, Cl, or $NTf_2$, 15 equiv.) in 0.5 mL $CH_3OH$, a large amount of precipitate appeared immediately. The precipitate was collected by filtration, washed with $H_2O$ (5 mL for X = Cl, 3 × 10 mL for others), and dried under vacuum to obtain the corresponding compounds (>80% yield).

### Measurements

NMR spectra were recorded using a Bruker 400 MHz Avance III HD Smart Probe ($^1H$, $^{13}C$, $^{19}F$, and 2D experiments). Chemical shifts for $^1H$, $^{13}C$, and $^{19}F$ are reported in ppm on the $\delta$ scale; $^1H$, $^{13}C$, and $^{19}F$ were referenced to the residual solvent peak. Coupling constants ($J$) are reported in Hz. UV-vis absorption spectroscopy was recorded on a HIMADZU UV-2700 spectrometer in the 200–800 nm regions. Electron-spin resonance (ESR) signals were recorded on a Bruker Elexsys 580 spectrometer with a 100 kHz magnetic field in the X band at room temperature. Photochromic experiments were carried out in CEL-HXUV300 300 W xenon lamp system. All anaerobic operations are carried out using the LG1200/750TS glovebox. Photocatalysis was performed in Photosyn-10 parallel photoreactor, equipped with a 400 nm LED light. High-resolution mass spectra were collected on an HPLC-Q-TOF-MS spectrometer in acetonitrile/methanol solution. Elemental analyses were performed on an Elementar Vario ELIII analyzer. Anion contents were measured using an ICS-5000+/900 ion chromatograph.

## Data availability

The authors declare that the data supporting this study are available within the paper and its supplementary information file. Additional data are available from the corresponding authors upon request. Correspondence and requests for materials should be addressed to D.Z. or E.-Q.G. Crystallographic data for the structures reported in this paper have been deposited at the Cambridge Crystallographic Data Centre (CCDC) under the deposition numbers 2356395, **1**-SCN; 2356397, **1**-I; 2356396, **1**-Br; 2357282, **1'**-SCN; 2359410, **1'**-Br; and 2356398, **3**-SCN. Copies of these data can be obtained free of charge via www.ccdc.cam.ac.uk/data_request/cif.

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

## Acknowledgements

This work was supported by the National Natural Science Foundation of China (22201075, D.Z.; 21971069, E.-Q.G.). R.L. is a Postdoctoral Researcher of the Fonds de la Recherche Scientifique—FNRS. D.Z., E.-Q.G., and S.L. are grateful for financial support from the SINOPEC Research Institute of Petroleum Processing.

## Author contributions

D.Z., E.-Q.G., and G.L. conceived and designed the research. G.L. carried out the majority of the experimental work. Z.D. synthesized cage **3**. Y.L. and Y.X. synthesized some tetraarylborate guests. C.W., R.L., and S.L. contributed to the analysis and interpretation of the results. G.L. wrote the initial draft of the manuscript. All authors edited the manuscript.

## Competing interests

The authors declare no competing interests.
