## [Transparent Peer Review file · Nature Communications]

Charge-Transfer Complexation of Coordination Cages for Enhanced Photochromism and Photocatalysis

Corresponding Author: Professor Dawei Zhang

Version 0:

Reviewer comments:

Reviewer #1

(Remarks to the Author)

The work by Li et al. describes a new synthetic strategy for preparing soluble Zr-based metal-organic cages (MOCs), highlighting their excellent performance in photochromism and photocatalysis. BARF- counter anions are utilized to enhance the solubility of Zr-MOCs, enabling the preparation of new cages in various shapes in solution. By incorporating multiple electron-deficient pyridinium groups into a single cage, the Zr-MOCs exhibit intensified positive electrostatic fields and electron-accepting abilities, capable of hosting electron-donating anions through charge-transfer (CT) interactions. The authors demonstrate that CT interactions between host and guest molecules enable efficient photoinduced electron transfer (PET), enhancing both photochromic properties and photocatalytic efficiency of Zr-MOCs. The approach in this study opens avenues for exploring solution chemistry of Zr-MOCs, with the strategy's reliability well-demonstrated. The enhanced behaviors in photochromism and photocatalysis underscore the potential of CT-mediated PET applications. Crystal structures, including those of the host alone, host-guest complexes, and their radical states, are thoroughly described, substantiating the findings. The experiments are carefully designed and conducted, and the Supporting Information is comprehensive. Overall, this original and compelling work merits publication in Nat. Commun. Minor revisions are recommended:

1. Ensure consistency between the manuscript title and that in the Supporting Information.
2. There are eight figures in the manuscript, including three small figures depicting crystal structures. Combining all crystal structures into a single figure may enhance the manuscript's clarity and organization.
3. The authors describe the synthesis of 1-Cl through precipitation, and the precipitate was collected after filtration and washing with H₂O. However, the ¹H NMR spectrum of 1-Cl in Figure S23 was given in D₂O. The authors should provide an explanation for this discrepancy.
4. In Figure S32, the ¹H NMR spectra of cage 2-BARF show equilibrium species between the ligand the cage. Is there any anion template effect to shift the equilibrium towards the cage?
5. The authors should clarify why long aliphatic substituent groups were introduced into the Cp₃Zr₃O(OH)₃ vertices of cage 3.

Reviewer #2

(Remarks to the Author)

The manuscript by Tao, Zhang, and coworkers presents the design and synthesis of a series of cationic coordination cages for anion binding and photocatalysis. They show that by using salts with higher solubility, they can prepare cages with higher solubility (this part is not very exciting or novel). The subsequent anion binding is interesting and well done, as is the photochemistry. After clarifying some minor issues, the manuscript could be accepted for publication in Nature Communications.

Specific comments:

- 1) The authors play up the soluble cage aspect a bit too much. Many researchers in this field (and dozens of other fields) routinely use anion exchange to prepare soluble products. A recent manuscript by Cook highlights the use of soluble starting materials (triflate salts) to make soluble cages. Chloride to triflate anions can easily be done with AgOTf reactions after product synthesis. Many researchers in this area have done this. I suggest the authors rewrite this part of the manuscript.

- 2) The authors argue that they use BArF salts to make soluble cages on account of the anion, however, they show in their mass spec (and a couple other spots) that the product they actually get is typically a mixture of BArF and chloride, they should clearly disclose this.
- 3) They don't really specifically give the composition of the cages. They just write: 1-anion without specifying the exact composition.
- 4) They state "It represents the first example of helical Zr-MOCs". This is not true.
- 5) There are some papers reporting anion binding with Zr cages (Synthesis, crystal structure and iodine capture of Zr-based metal-organic polyhedron). Please discuss this.
- 6) Tim Cook has shown (in either a Dalton or Chem. Commun. paper) that these cages can be solubilized by functionalizing the Cp groups on the cage. They have presumably used this strategy here but don't really talk about it.
- 7) The crystal structures could use some closer inspection. Where are the rest of the anions? The number of anions in the supporting info (1-SCN for example has 4 SCN groups in the structure but the SI says there are 8) don't match the structures.

Overall, this is a nice paper, just too much up front claiming to invent a new way for making these types of cages.

Reviewer #3

(Remarks to the Author)

The manuscript by Gao, Zhang and coworkers describes the synthetic access to novel coordination cages, with some applications in photocatalysis.

The study of the formation of these supramolecular assemblies and their characterization has been done thoroughly, with evaluation of diverse counter-anions through titration and/or NMR. These compounds have shown to undergo sensitization under light irradiation, and the authors have proposed some mechanisms for electron transfer between the different entities of the molecular systems and O₂.

Remark, Fig. 1: the stoichiometry between the different elements is not very clear.

Different MOCs were further evaluated as sensitizers for aryl-coupling from organoborates. Although the authors claim these systems to be widely applicable, this is the only application described here. One of the last parts focuses then on this C-C bond formation, applying these newly accessed MOCs as catalysts for the photooxidation of NaBPh₄. For this purpose, they used 1-NTf₂ and conducted optimizations up to 1-NO₃ (Table 1). Two of these MOCs show great activity for the formation of Ph-Ph. However, 1-BPh₄ was not tested in this photooxidation process, and there is no comment associated to it within the text. In the studies conducted in Fig. 5, did the authors observed some coupling products? As BPh₄⁻ is the counter-anion, there should be – at least – some residual Ph-Ph in solution.

Other comment: There are major inconsistencies in references. A full list of authors is sometimes give, other times only first author et. al. This should be corrected.

Version 1:

Reviewer comments:

Reviewer #1

(Remarks to the Author)

I think the authors addressed well my comments and so the paper can be published now.

I have carefully reviewed the reports from reviewers 2 and 3, as well as the authors' responses. I am pleased to say that the authors have made substantial improvements to their manuscript based on the feedback received.

I believe these revisions are commendable and address the concerns raised by the reviewers effectively. Therefore, I recommend accepting the article for publication in Nature Communications.

Point by point response to the reviewers:

Referee: 1

The work by Li *et al.* describes a new synthetic strategy for preparing soluble Zr-based metal-organic cages (MOCs), highlighting their excellent performance in photochromism and photocatalysis. BAr_F^- counter anions are utilized to enhance the solubility of Zr-MOCs, enabling the preparation of new cages in various shapes in solution. By incorporating multiple electron-deficient pyridinium groups into a single cage, the Zr-MOCs exhibit intensified positive electrostatic fields and electron-accepting abilities, capable of hosting electron-donating anions through charge-transfer (CT) interactions. The authors demonstrate that CT interactions between host and guest molecules enable efficient photoinduced electron transfer (PET), enhancing both photochromic properties and photocatalytic efficiency of Zr-MOCs. The approach in this study opens avenues for exploring solution chemistry of Zr-MOCs, with the strategy's reliability well-demonstrated. The enhanced behaviors in photochromism and photocatalysis underscore the potential of CT-mediated PET applications. Crystal structures, including those of the host alone, host-guest complexes, and their radical states, are thoroughly described, substantiating the findings. The experiments are carefully designed and conducted, and the Supporting Information is comprehensive. Overall, this original and compelling work merits publication in *Nat. Commun.* Minor revisions are recommended:

1. Ensure consistency between the manuscript title and that in the Supporting Information.

Reply: We apologize for the oversight in our initial submission. The title in the revised Supporting Information has been corrected to be consistent with that in the main MS.

Supplementary Information, Page S1:

Supplementary Information for

Charge-Transfer Complexation of Coordination Cages for Enhanced Photochromism and Photocatalysis

Gen Li,¹ Zelin Du,¹ Chao Wu,² Yawei Liu,¹ Yan Xu,¹ Roy Lavendomme,^{3,4} En-Qing Gao,^{1,5*} and Dawei Zhang^{1,5*}

2. There are eight figures in the manuscript, including three small figures depicting crystal structures. Combining all crystal structures into a single figure may enhance the manuscript's clarity and organization.

Reply: We agree with the reviewer that combining all figures depicting crystal structures into a single figure can make the manuscript neater and clearer. We have thus combined the original **Figures 3 and 6**, which show the ground-state and radical host-guest crystal structures, into a single figure (**Figure 3** in the revised manuscript). This change can not only improve the manuscript's clarity but also facilitates easier comparison between the original and radical-state host-guest structures. However, after careful consideration, we have decided not to combine the original **Figure 2** into the same figure, as it only shows the crystal structures of the hosts.

MS, Page 9:

Fig. 3 | X-ray crystallographic structures of host-guest complexes. (a) Structure of $I^- \subset 1$ in **1-I**. (b) Structure of $Br^- \subset 1$ in **1-Br**. (c) Structure of $Br^- \subset 1^{\bullet}$ in **1-Br** and the structure parameters of the three viologen units showing the radical state of the cage.

3. The authors describe the synthesis of **1-Cl** through precipitation, and the precipitate was collected after filtration and washing with H_2O . However, the 1H NMR spectrum of **1-Cl** in Figure S23 was given in D_2O . The authors should provide an explanation for this discrepancy.

Reply: We appreciate the reviewer for pointing out this discrepancy. Complex **1-Cl** was precipitated by mixing **1-NO₃** and TBACl in water and was obtained after filtration. To remove the residual TBACl attached on the solid, water was used to wash the sample. **1-Cl** is modestly soluble in water, so only a small amount of water (5 mL) was used for washing. The modest solubility allows its measurement of NMR spectroscopy in D_2O . To avoid the discrepancy mentioned by the reviewer, we have added explanation into the revised Supplementary Information.

Supplementary Information, Page S14:

1-X (X = OTf, PF₆, Cl, and NTf₂): **1-NO₃** (50 mg, 17 μ mol) was dissolved in 5.0 mL H_2O . Upon addition of TBAX (X = OTf, PF₆, Cl, or NTf₂, 15 equiv.) in 0.5 mL CH_3OH , a large amount of precipitate appeared immediately. For **1-X** (X = OTf, PF₆, and NTf₂), the precipitate was collected after filtration and washed with H_2O (3×10 mL), and was dried under vacuum to obtain **1-X** (> 80% yield). **1-Cl** was collected in a similar way (~ 80% yield), but a smaller amount of water (5 mL) was used for washing because **1-Cl** is modestly soluble in water. The simplified washing for **1-Cl** is sufficient for removing the attached TBACl due to the high solubility of TBACl in water.

4. In Figure S32, the ^1H NMR spectra of cage **2**-BAR_F show equilibrium species between the ligand and the cage. Is there any anion template effect to shift the equilibrium towards the cage?

Reply: Following the suggestion from the reviewer and based upon the anion binding properties of **1**-BAR_F, we tested the anion template effect on the formation of cage **2**. We observed that the addition of other anions, such as I^- or TfO^- , into the solution of **2**-BAR_F shifted the equilibrium towards the cage. These results have been added into the revised MS and Supplementary Information.

MS, Page 7:

2-BAR_F ($\{[\text{Cp}_3\text{Zr}_3(\mu_3\text{-O})(\mu_2\text{-OH})_3]_2(\text{L}^2)_3\}(\text{BAR}_F)_8$) was found to partially dissociate into ligands and Zr-clusters in CD_3OD , $\text{DMSO-}d_6$, and CD_3COCD_3 , while negligible dissociation occurred in CD_3CN (Supplementary Fig. 32). The presence of additional anions, such as I^- and TfO^- , could also serve as templates to drive the conversion from the assembly components into the cage (Supplementary Figs. 33-34).

Supplementary Information, Page S21:

Supplementary Fig. 33 ^1H NMR spectra (CD_3OD , 400 MHz, 298 K) of **2**-BAR_F in the presence of varying equivalents of I^- , showing the template effect of I^- that drives the formation of the cage.

Supplementary Information, Page S22:

Supplementary Fig. 34 ^1H NMR spectra (CD_3OD , 400 MHz, 298 K) of **2-BAr_F** in the presence of varying equivalents of TfO^- , showing the template effect of TfO^- that drives the formation of the cage.

5. The authors should clarify why long aliphatic substituent groups were introduced into the $\text{Cp}_3\text{Zr}_3\text{O}(\text{OH})_3$ vertices of cage **3**.

Reply: We appreciate the reviewer's suggestion. We observed significant dissociation of the cage when using conventional Cp_2ZrCl_2 for self-assembly with the thiazolo[5,4-d]thiazole-extended viologen ligand. However, the introduction of butyl groups into the $\text{Cp}_3\text{Zr}_3\text{O}(\text{OH})_3$ vertices led to the complete formation of **3-BAr_F** with negligible dissociation. This improvement is presumably due to enhanced solubility and stability of the cage. Following the reviewer's suggestion, we have included this explanation in the revised MS.

MS, Page 7:

The assembly of a thiazolo[5,4-*d*]thiazole-extended viologen ligand with dibutylzirconocene dichloride ($(n\text{-BuCp})_2\text{ZrCl}_2$) led to an elongated C_2L_3 helicate (**3-BAr_F**, $\{[(n\text{-BuCp})_3\text{Zr}_3(\mu_3\text{-O})(\mu_2\text{-OH})_3]_2(\text{L}^3)_3\}(\text{BAr}_\text{F})_8$) (Supplementary Figs. 35-41). We noted that the assembly using Cp_2ZrCl_2 showed significant dissociation in CD_3OD , while the use of $(n\text{-BuCp})_2\text{ZrCl}_2$ enabled the formation of **3** with negligible dissociation. This could be because the *n*-butyl groups decorating $\text{Cp}_3\text{Zr}_3\text{O}(\text{OH})_3$ vertices further increase the solubility and stability of the cage in organic solvents.⁵³

MS, Page 29:

53. Sullivan, M.G. et al. Altering the solubility of metal-organic polyhedra via pendant functionalization of $\text{Cp}_3\text{Zr}_3\text{OOH}_3$ nodes. *Dalton Trans.* **52**, 338-346 (2023).

Referee: 2

The manuscript by Gao, Zhang, and coworkers presents the design and synthesis of a series of cationic coordination cages for anion binding and photocatalysis. They show that by using salts with higher solubility, they can prepare cages with higher solubility (this part is not very exciting or novel). The subsequent anion binding is interesting and well done, as is the photochemistry. After clarifying some minor issues, the manuscript could be accepted for publication in *Nature Communications*.

1. The authors play up the soluble cage aspect a bit too much. Many researchers in this field (and dozens of other fields) routinely use anion exchange to prepare soluble products. A recent manuscript by Cook highlights the use of soluble starting materials (triflate salts) to make soluble cages. Chloride to triflate anions can easily be done with AgOTf reactions after product synthesis. Many researchers in this area have done this. I suggest the authors rewrite this part of the manuscript.

Reply: We greatly appreciate the very insightful comment from the reviewer. After perceiving the comment, we have carefully considered whether our initial description on the synthesis of soluble Zr-MOCs was appropriate or not. We recognize and fully agree with the reviewer that the earlier statements were overstated. In our initial description, we overemphasized the novelty of the method for synthesizing high-solubility Zr-MOCs. However, as the reviewer noted in this and other comments, introducing solubilizing groups into the ligands or vertices can enhance the solubility of Zr-MOCs, and anion exchange after cage formation is also a common method. These suggestions prompted us to delve deeper into the differences between our method and the previous methods, in particular for the traditional anion exchange strategy. We noted that the previous work typically involves anion exchange after obtaining the cage, which requires Zr-MOCs to first crystallize using the traditional synthetic method. In contrast, our strategy utilizes solubilizing BARF^- as counteranions of ionic ligands, allowing the entire self-assembly process to remain homogeneous, which facilitates reversible self-assembly and self-correction.

Accordingly, following the suggestion from the reviewer, we have made substantial changes to our presentation, especially in the sections of Abstract, Introduction, and Conclusion. On one hand, we have systematically summarized methods for enhancing Zr-MOC solubility; On the other hand, we have clarified the differences and relationship between our method and the previous approaches. The goal of these revisions is to offer a more objective representation of the current state of high-solubility Zr-MOCs and the contribution of this work to the field.

MS, Page 1:

Abstract

Intensified host-guest electronic interplay within stable metal-organic cages (MOCs) present great opportunities for applications in stimuli response and photocatalysis. Zr-MOCs represent a type of robust discrete hosts for such a design, but their host-guest chemistry in solution is hampered by the limited solubility. Here, by using pyridinium-derived cationic ligands with tetrakis(3,5-bis(trifluoromethyl)phenyl)borate (BARF^-) as solubilizing counteranions (we describe an anion-assisted synthetic strategy for soluble Zr-MOCs), we report the preparation of soluble Zr-MOCs of different shapes (**1-4**) that are otherwise inaccessible through a conventional method. Enforced arrangement of the multiple electron-deficient pyridinium groups into one cage (**1**) leads to magnified positive electrostatic field and electron-accepting strength in favor of hosting electron-donating anions, including halides and tetraarylborates. The strong charge-transfer (CT) interactions

activate guest-to-host photoinduced electron transfer (PET), leading to pronounced and regulable photochromisms. Both ground-state and radical structures of host and host-guest complexes have been unambiguously characterized by X-ray crystallography. The CT-enhanced PET also enables the use of **1** as an efficient photocatalyst for aerobic oxidation of tetraarylborates into biaryls and phenols. This work presents the solution assembly of soluble Zr-MOCs from cationic ligands with the assistance of solubilizing anions (~~establishes an anion-assisted method for facile preparation of soluble Zr-MOCs~~) and highlights the great potential of harnessing host-guest CT for boosting PET-based functions and applications.

MS, Page 3:

The Zr-MOCs with $\text{Cp}_3\text{Zr}_3\text{O}(\text{OH})_3$ vertices ($\text{Cp} = \eta^5\text{-C}_5\text{H}_5$) and multicarboxylic linkers, which exhibit high chemical stability due to the high Zr–O bond energy (766 kJ/mol), have witnessed rapid development since the first report in 2013 by the Yuan group.^{43,44} They are usually synthesized through a conventional solvothermal approach to yield crystalline powders. The majority of Zr-MOCs have limited solubility resulting from the strong interactions between cationic $\text{Cp}_3\text{Zr}_3\text{O}(\text{OH})_3$ vertices and counterions (generally Cl^-), and are primarily used as crystalline solids for multiphase applications, such as gas separation⁴⁴⁻⁴⁶; iodine capture,⁴⁷ and heterogeneous catalysis.^{48,49} Nevertheless, a few Zr-MOCs have been investigated as hosts for anion binding in solution or processed in solution to prepare composite membranes.⁵⁰⁻⁵⁵ Improvement of the solubility of Zr-MOCs can be achieved by functionalizing the linkers with amino, alkyl, or other solubilizing groups.⁵⁰⁻⁵² Another rare but innovative method involves the decoration of the $\text{Cp}_3\text{Zr}_3\text{O}(\text{OH})_3$ nodes by introducing pendant groups, such as *n*-butyl, benzyl, or trifluoromethylbenzyl, into the Cp rings.^{53,54} A third solubilization strategy is postassembly ion exchange, which has been effective for various ionic MOCs including Zr-MOCs.^{55,56}

In this work, we highlight the use of counteranions to facilitate synthesis of highly soluble Zr-MOCs from pyridinium-derived cationic ligands. Different from the previous postassembly anion-exchange method,⁵⁵ we introduce the solubilizing counteranion, tetrakis(3,5-bis(trifluoromethyl)phenyl)borate (BAR_F^-), into the flexible pyridinium-based ligands to minimize cation-anion interactions during the self-assembly, thereby preventing the formation of insoluble intermediates. Following this approach, four Zr-MOCs ranging from helicates (**1-3**) to a tetrahedron (**4**) were constructed homogeneously in solutions. The incorporation of multiple electron-deficient pyridinium groups imparts superior electron-accepting abilities to the MOCs, which thus are capable of hosting electron-donating anions, including tetraarylborates, through dominative CT interactions. The ground-state interactions facilitates guest-to-host PET, resulting in efficient and regulable photochromism. In particular, the photogenerated radical state of **1** can rapidly transfer electrons to O_2 to generate $\text{O}_2^{\cdot-}$, enabling the use of **1** as an effective photocatalyst for oxidation of tetraarylborate guests.

MS, Page 23:

Discussion

In conclusion, (we have reported a new strategy for the synthesis of a series of Zr-MOCs.) introduction of BAR_F^- as the counteranions of cationic ligands enables homogeneous assembly processes in solutions, resulting in soluble Zr-MOCs. Helicate-like cage **1** is capable of binding a variety of anions, which is enabled by the Coulombic interactions for anions with poor ED strength and dominated by CT complexation for electron-donating halides and tetraarylborates. In comparison to the free ligand, the CT complexation is significantly enhanced owing to the cooperation of the multiple cationic viologen groups

in the cage. The enhanced binding leads to pronounced photochromism due to anion-to-viologen PET, which can be modulated by coexistent anions and was characterized by X-ray crystal structures. The photogenerated viologen radical readily undergoes ET to dioxygen to produce superoxide. Combining the efficient PET and superoxide-generation properties, **1** can effectively catalyze photooxidation of tetraarylborates, which allows the simultaneous synthesis of biaryls and phenols through aryl coupling and hydroxylation. This work (the validity of the anion-assisted synthetic strategy for soluble Zr-MOCs) demonstrates the great potential of using host-guest CT complexation of MOCs to manipulate PET for sophisticated responsive and catalytic performances. Extended exploration with various electron-deficient soluble Zr-MOCs (e.g., **2-4**) are underway in our laboratory.

MS, Pages 28-29:

47. Cheng, S. et al. Synthesis, crystal structure and iodine capture of Zr-based metal-organic polyhedron. *Inorg. Chim. Acta* **516**, 120174 (2021).

51. Shi, W.J. et al. Supramolecular coordination cages based on N-heterocyclic carbene-gold(I) ligands and their precursors: self-assembly, structural transformation and guest-binding properties. *Chem. Eur. J.* **27**, 7853-7861 (2021).

52. Liu, J. et al. Self-healing hyper-cross-linked metal-organic polyhedra (HCMOPs) membranes with antimicrobial activity and highly selective separation properties. *J. Am. Chem. Soc.* **141**, 12064-12070 (2019).

53. Sullivan, M.G. et al. Altering the solubility of metal-organic polyhedra via pendant functionalization of $Cp_3Zr_3OOH_3$ nodes. *Dalton Trans.* **52**, 338-346 (2023).

54. Li, Y. et al. Artificial biomolecular channels: enantioselective transmembrane transport of amino acids mediated by homochiral zirconium metal-organic cages. *J. Am. Chem. Soc.* **143**, 20939-20951 (2021).

55. Pastore, V.J. et al. Clickable norbornene-based zirconium carboxylate polyhedra. *Chem. Mater.* **35**, 1651-1658 (2023).

56. Grommet, A.B. et al. Anion exchange drives reversible phase transfer of coordination cages and their cargoes. *J. Am. Chem. Soc.* **140**, 14770-14776 (2018).

2. The authors argue that they use BArF salts to make soluble cages on account of the anion, however, they show in their mass spec (and a couple other spots) that the product they actually get is typically a mixture of BArF and chloride, they should clearly disclose this.

Reply: We thank the reviewer for pointing this out. As initially presented in the caption of the figure, the high-resolution ESI-mass spectrometry analysis of **1**-BArF was conducted in the presence of TBACl. No reasonable signals could be detected for **1**-BArF in the absence of TBACl, presumably due to the lower stability of **1** under MS conditions without a suitable anion template. We have added this explanation into the revised caption of Supplementary Fig. 22.

Supplementary Information, Page S14:

Supplementary Fig. 22 | High-resolution ESI-mass spectrometry analysis of **1**-BArF in the presence of TBACl showing the 3+ and 4+ peaks. No reasonable signals were detected for

1-BAr_F in the absence of TBACl, presumably due to the lower stability of **1** under MS conditions without a suitable anion template.

3. They don't really specifically give the composition of the cages. They just write: **1**-anion without specifying the exact composition.

Reply: We agree with the reviewer that the information on the composition of the cage is important. Following the suggestion, the formulas of cages are given in parentheses in the revised MS.

MS, Page 5:

Fig. 1 Self-assembly of metal-organic cages. **a**, Self-assembly of **1-BAr_F** (C₂L₃, {[Cp₃Zr₃(μ₃-O)(μ₂-OH)₃]₂(L¹)₃}(BAr_F)₈). **b**, ¹H NMR spectrum (CD₃OD, 400 MHz, 298K) of **1-BAr_F**. **c**, Structures of **2-BAr_F** (C₂L₃, {[Cp₃Zr₃(μ₃-O)(μ₂-OH)₃]₂(L²)₃}(BAr_F)₈), **3-BAr_F** (C₂L₃, {[*n*-BuCp)₃Zr₃(μ₃-O)(μ₂-OH)₃]₂(L³)₃}(BAr_F)₈), and **4-BAr_F** (C₄L₄, {[Cp₃Zr₃(μ₃-O)(μ₂-OH)₃]₄(L⁴)₄}(BAr_F)₁₆).

MS, Page 6:

As shown in Fig. 1a, **1-BAr_F** ([Cp₃Zr₃(μ₃-O)(μ₂-OH)₃]₂(L¹)₃}(BAr_F)₈) was assembled from a 1:2 ratio of **L¹-BAr_F** and Cp₂ZrCl₂ in CH₃OH/H₂O at 65 °C for 12 h. The reaction gave a homogeneous solution and **1-BAr_F** was precipitated by adding a large amount of water. The identity of the cage was confirmed by high-resolution electrospray-ionization mass spectrometry (HR-ESI-MS), which is consistent with a C₂L₃ [C = cluster Cp₃Zr₃(μ₃-O)(μ₂-OH)₃, and L = Ligand] composition (Supplementary Fig. 22).

MS, Page 6:

The counteranions can be easily exchanged to other anions to obtain **1-X** complexes ([Cp₃Zr₃(μ₃-O)(μ₂-OH)₃]₂(L¹)₃}(X)₈) with X = Tf₂N⁻ (bis(trifluoromethanesulfonyl)imide), TfO⁻ (trifluoromethanesulfonate), PF₆⁻, Cl⁻, or NO₃⁻ (Supplementary Figs. 23-24).

MS, Page 7:

Cage **2** is isomeric to **1**, with the difference being in the carboxylate position in the ligands (Supplementary Figs. 25-32). **2-BAr_F** ([Cp₃Zr₃(μ₃-O)(μ₂-OH)₃]₂(L²)₃}(BAr_F)₈) was found to partially dissociate into ligands and Zr-clusters in CD₃OD, DMSO-*d*₆, and CD₃COCD₃, while negligible dissociation occurred in CD₃CN (Supplementary Fig. 32). The presence of additional anions, such as I⁻ and TfO⁻, could also serve as templates to drive the conversion from the assembly components into the cage (Supplementary Figs. 33-34). The assembly of a thiazolo[5,4-*d*]thiazole-extended viologen ligand with dibutylzirconocene dichloride ((*n*-BuCp)₂ZrCl₂) led to an elongated C₂L₃ helicate (**3-BAr_F**, [(*n*-BuCp)₃Zr₃(μ₃-O)(μ₂-OH)₃]₂(L³)₃}(BAr_F)₈) (Supplementary Figs. 35-41). We noted that the assembly using Cp₂ZrCl₂ showed significant dissociation in CD₃OD, while the use of (*n*-BuCp)₂ZrCl₂ enabled the formation of **3** with negligible dissociation. This could be because the *n*-butyl groups decorating Cp₃Zr₃O(OH)₃ vertices further increase the solubility and stability of the cage in organic solvents.⁵³ A face-capped C₄L₄ tetrahedral cage **4-BAr_F** ([Cp₃Zr₃(μ₃-O)(μ₂-OH)₃]₄(L⁴)₄}(BAr_F)₁₆) was also successfully assembled from a tripyridinium-tricarboxylate ligand and Cp₂ZrCl₂ (Supplementary Figs. 42-48).

4. They state "It represents the first example of helical Zr-MOCs". This is not true.

Reply: Following the suggestion from the reviewer, this statement has been removed in the revised MS to be more precise.

MS, Page 6:

As shown in Fig. 2a, MOC 1 is a cage-like triple helicate with two trinuclear $[\text{Cp}_3\text{Zr}_3(\mu_3\text{-O})(\mu_2\text{-OH})_3]$ clusters connected by three viologen-based carboxylate linkers. (It represents the first example of helical Zr-MOCs.)

5. There are some papers reporting anion binding with Zr cages (Synthesis, crystal structure and iodine capture of Zr-based metal-organic polyhedron). Please discuss this.

Reply: We apologize for missing out the interesting and relevant literature on Zr-MOCs. The specific work mentioned by the reviewer (*Inorg. Chim. Acta* **516**, 120174 (2021)), which reports on iodine capture using solid-state Zr-MOCs, has been discussed and cited in the revised MS. In addition, a paper focusing on the binding of sulfonate anions with a Zr-MOC (*Chem. Eur. J.* **27**, 7853-7861 (2021)) has also been cited and highlighted.

MS, Page 3:

The majority of Zr-MOCs have limited solubility resulting from the strong interactions between cationic $\text{Cp}_3\text{Zr}_3\text{O}(\text{OH})_3$ vertices and counterions (generally Cl^-), and are primarily used as crystalline solids for multiphase applications, such as gas separation⁴⁴⁻⁴⁶ iodine capture,⁴⁷ and heterogeneous catalysis.^{48, 49} Nevertheless, a few Zr-MOCs have been investigated as hosts for anion binding in solution or processed in solution to prepare membranes.⁵⁰⁻⁵⁵

MS, Pages 28-29:

47. Cheng, S. et al. Synthesis, crystal structure and iodine capture of Zr-based metal-organic polyhedron. *Inorg. Chim. Acta* **516**, 120174 (2021).

51. Shi, W.J. et al. Supramolecular coordination cages based on N-heterocyclic carbene-gold(I) ligands and their precursors: self-assembly, structural transformation and guest-binding properties. *Chem. Eur. J.* **27**, 7853-7861 (2021).

6. Tim Cook has shown (in either a *Dalton* or *Chem. Commun.* paper) that these cages can be solubilized by functionalizing the Cp groups on the cage. They have presumably used this strategy here but don't really talk about it.

Reply: We are very grateful to the reviewer for pointing out this very important point. The paper by Cook about solubilizing the cages through functionalizing the Cp groups (*Dalton Trans.* **52**, 338-346 (2023)) has been cited and described in the revised MS together with other work involving the strategy. In addition, we have also referenced the work in the context of synthesizing **3**-BAR_F with long aliphatic chains on Cp rings.

MS, Page 3:

Nevertheless, a few Zr-MOCs have been investigated as hosts for anion binding in solution or processed in solution to prepare composite membranes.⁵⁰⁻⁵⁵ Improvement of the solubility of Zr-MOCs can be achieved by functionalizing the linkers with amino, alkyl, or other solubilizing groups.⁵⁰⁻⁵² Another rare but innovative method involves the decoration of the $\text{Cp}_3\text{Zr}_3\text{O}(\text{OH})_3$ nodes by introducing pendant groups, such as *n*-butyl, benzyl, or trifluoromethylbenzyl, into the Cp rings.^{53, 54}

MS, Page 7:

We noted that the assembly using Cp_2ZrCl_2 showed significant dissociation in CD_3OD , while the use of $(n\text{-BuCp})_2\text{ZrCl}_2$ enabled the formation of **3** with negligible dissociation. This could be because the *n*-butyl groups decorating $\text{Cp}_3\text{Zr}_3\text{O}(\text{OH})_3$ vertices further increase the solubility and stability of the cage in organic solvents.⁵³

MS, Page 29:

53. Sullivan, M.G. et al. Altering the solubility of metal-organic polyhedra via pendant functionalization of $\text{Cp}_3\text{Zr}_3\text{OOH}_3$ nodes. *Dalton Trans.* **52**, 338-346 (2023).

7. The crystal structures could use some closer inspection. Where are the rest of the anions? The number of anions in the supporting info (**1**-SCN for example has 4 SCN groups in the structure but the SI says there are 8) don't match the structures.

Reply: We appreciate the reviewer for pointing out this. For clarity, only the anions inside the cages are shown in the figures of the main MS. The rest of the anions are outside the cages. To illustrate this, we have included additional figures (Supplementary Figs. 49-50) in the revised Supplementary Information, where the anions locating both inside and outside the cages are shown in different colors.

As the nonradical cages each possess eight positive charges, there are eight counteranions per cage, which is confirmed by NMR for **1**-BAR_F and ion chromatography for **1**-SCN and **1**-Br. However, in the crystal structures of **1**-SCN and **3**-SCN, only four and three anions were located, respectively, with none inside the cage. The other anions could not be located due to the high disorder of the anions and the limited quality of the data. For similar reasons, we could locate only two SCN⁻ anions for **1**[•]-SCN, which contains six SCN⁻ anions according to ion chromatography. We have thus provided additional statements in section 3 of the revised Supplementary Information to explain the mismatch in anion number. All eight anions in the crystal structures of **1**-I and **1**-Br could be located. Seven anions are located in **1**[•]-Br, in agreement with the measurement of ion chromatography.

Supplementary Information, Page S33:

Supplementary Fig. 49 | Views showing the cages and anions in **1**-I, **1**-Br and **1**[•]-Br. Dark red and orange balls represent the anions located inside and outside the cage, respectively. The dashed lines represent the interactions of the anions with the cages.

Supplementary Information, Page S33:

Supplementary Fig. 50 1 Views showing the cages and anions in **1-SCN**, **3-SCN** and **1*-SCN**. All anions are outside the cages. The dashed lines represent the interactions of the anions with the cages.

Supplementary Information, Page S31:

1-SCN: 1·8SCN [+ solvent], CCDC 2356395

Formula $C_{116}H_{96}N_{14}O_{20}S_8Zr_6$, M 2809.95, Monoclinic, space group $P2_1/c$ (#14), a 22.5191(5), b 16.2430(4), c 43.8728(1) Å, β 98.7080(1)°, V 15862.7(7) Å³, D_c 1.079 g cm⁻³, Z 4, crystal size 0.32 by 0.24 by 0.20 mm, colour pale yellow, block, temperature 173(2) K, λ (Cu-K α) 1.54178 Å, $2\theta_{max}$ 136.88, $R1(F)$ 0.0499, $wR2(F^2)$ 0.1408, GoF(all) 1.061, μ (mm⁻¹) 4.025, $F(000)$ 5208.

Note: Ion chromatography revealed that there are eight SCN⁻ per cage for **1-SCN**, which is as expected for the charge balance of **1⁸⁺**. Only four SCN⁻ anions (all outside the cage) per cage were located by X-ray crystallography, and the remaining four SCN⁻ anions could not be located perhaps due to the high disorder of the anions and the limited quality of the data.

Supplementary Information, Page S32:

3-SCN: 3·8SCN [+ solvent], CCDC 2356398

Formula $C_{152}H_{144}N_{20}O_{20}S_{14}Zr_6$, M 3567.13, Monoclinic, space group $P2_1/c$ (#14), a 26.4846(7), b 26.4962(7), c 31.5550(8) Å, β 109.7510(10)°, V 20840.7(9) Å³, D_c 1.018 g cm⁻³, Z 4, crystal size 0.32 by 0.25 by 0.16 mm, colour yellow, block, temperature 173(2) K, λ (Cu-K α) 1.54178 Å, $2\theta_{max}$ 136.48, $R1(F)$ 0.0660, $wR2(F^2)$ 0.2242, GoF(all) 1.046, μ (mm⁻¹) 3.608, $F(000)$ 6508.

Note: Only three SCN⁻ anions (all outside the cage) per cage were located by X-ray crystallography, and the remaining five SCN⁻ anions could not be located perhaps due to the high disorder of the anions and the limited quality of the data.

Supplementary Information, Page S32:

1*-SCN: 1*·6SCN [+ solvent], CCDC 2357282

Formula $C_{114}H_{96}N_{12}O_{20}S_6Zr_6$, M 2693.79, Orthorhombic, space group $Aba2$ (#41), a 31.481(6), b 24.112(5), c 21.777(4) Å, V 16530(6) Å³, D_c 0.989 g cm⁻³, Z 4, crystal size

0.29 by 0.21 by 0.14 mm, colour blue, block, temperature 293(2) K, $\lambda(\text{Mo-K}\alpha)$ 0.71037 Å, $2\theta_{\text{max}}$ 50.82, $R1(F)$ 0.1089, $wR2(F^2)$ 0.2988, GoF(all) 1.247, $\mu(\text{mm}^{-1})$ 0.436, $F(000)$ 4976.

Note: Ion chromatography revealed that there are six SCN^- per cage for **1**-SCN. Only two SCN^- anions (both outside the cage) per cage were located by X-ray crystallography, and the remaining four SCN^- anions could not be located perhaps due to the high disorder of the anions and the limited quality of the data.

Overall, this is a nice paper, just too much up front claiming to invent a new way for making these types of cages.

Reply: We thank the Reviewer once again for all the very helpful suggestions. After addressing these issues, we feel the quality of the manuscript has been greatly improved. We welcome additional comments or suggestions from the reviewer and are happy to address them further to enhance the quality of this work.

Referee: 3

The manuscript by Gao, Zhang and coworkers describes the synthetic access to novel coordination cages, with some applications in photocatalysis.

The study of the formation of these supramolecular assemblies and their characterization has been done thoroughly, with evaluation of diverse counter-anions through titration and/or NMR. These compounds have shown to undergo sensitization under light irradiation, and the authors have proposed some mechanisms for electron transfer between the different entities of the molecular systems and O_2 .

1. Fig. 1: the stoichiometry between the different elements is not very clear.

Reply: We appreciate the suggestion from the reviewer, which has been also noted by reviewer 2. Following the suggestion, we have added the stoichiometry between the ligand and Zr-cluster for each cage in the revised caption of Fig. 1. The exact composition of each cage has been also added.

MS, Page 5:

Fig. 1 Self-assembly of metal-organic cages. **a**, Self-assembly of **1**- BAr_F (C_2L_3 , $\{[\text{Cp}_3\text{Zr}_3(\mu_3\text{-O})(\mu_2\text{-OH})_3]_2(\text{L}^1)_3\}(\text{BAr}_F)_8$). **b**, ^1H NMR spectrum (CD_3OD , 400 MHz, 298K) of **1**- BAr_F . **c**, Structures of **2**- BAr_F (C_2L_3 , $\{[\text{Cp}_3\text{Zr}_3(\mu_3\text{-O})(\mu_2\text{-OH})_3]_2(\text{L}^2)_3\}(\text{BAr}_F)_8$), **3**- BAr_F (C_2L_3 , $\{[(n\text{-BuCp})_3\text{Zr}_3(\mu_3\text{-O})(\mu_2\text{-OH})_3]_2(\text{L}^3)_3\}(\text{BAr}_F)_8$), and **4**- BAr_F (C_4L_4 , $\{[\text{Cp}_3\text{Zr}_3(\mu_3\text{-O})(\mu_2\text{-OH})_3]_4(\text{L}^4)_4\}(\text{BAr}_F)_{16}$).

2. Different MOCs were further evaluated as sensitizers for aryl-coupling from organoborates. Although the authors claim these systems to be widely applicable, this is the only application described here. One of the last parts focuses then on this C-C bond formation, applying these newly accessed MOCs as catalysts for the photooxidation of NaBPh_4 . For this purpose, they used **1**-NTf₂ and conducted optimizations up to **1**-NO₃ (Table 1). Two of these MOCs show great activity for the formation of Ph-Ph. However, **1**-BPh₄ was not tested in this photooxidation process, and there is no comment associated to it within the text. In the studies conducted in Fig. 5, did the authors observed some coupling products? As BPh_4^- is the counter-anion, there should be – at least – some residual Ph-Ph in solution.

Reply: As mentioned by the reviewer, we selected photocatalytic aryl-coupling from organoborates to demonstrate the application associated with the host-guest CT and PET properties of the viologen-

based cages. **1**-NTf₂ and **1**-NO₃ were chosen because (i) Tf₂N⁻ and NO₃⁻ are CT- and PET-inactive so that they do not perturb the photocatalytic processes; (ii) Tf₂N⁻ shows the weakest binding with **1** so that it would not perturb the interactions of **1** with organoborate substrates and thus the electron transfer from substrates to **1**. (iii) **1**-NO₃ is soluble in water and thus selected for studies in aqueous media. These points have been briefed in the main MS.

Following the suggestion from the reviewer, we prepared complex **1**-BPh₄ and tested its photocatalytic performance for substrate BPh₄⁻. It proved that the use of **1**-BPh₄ instead of **1**-NTf₂ is somewhat beneficial to photooxidation of BPh₄⁻ (the yield increased from 92% to 98%). This is consistent with the anion effect studied in the main MS: the absence of competing counteranions facilitates photooxidation to the most. The new result has been included in the revised Table 1 and commented in the revised MS. We did not use **1**-BPh₄ for further catalytic reactions of other tetraarylborates because its photooxidation products would contaminate the oxidation products of other substrates.

MS, Page 17:

Table 1. Photocatalytic transformation of BPh₄⁻.

Entry	Catalyst	Solvent	Atmos.	Conv. (%) ^[a]
1	1 -NTf ₂	CH ₃ OH	O ₂	92 (100 ^[b])
2	1 -NTf ₂	CH ₃ OH	air	81
3	/	CH ₃ OH	O ₂	trace
4 ^[c]	1 -NTf ₂	CH ₃ OH	O ₂	trace
5	1 -NTf ₂	CH ₃ OH	N ₂	6
6 ^[d]	Cp ₂ ZrCl ₂	CH ₃ OH	O ₂	trace
7 ^[e]	L ¹ -NTf ₂	CH ₃ OH	O ₂	31
8	1 -NTf ₂	CH ₃ COCH ₃	O ₂	43
9	1 -NTf ₂	DMSO	O ₂	51
10	1 -NTf ₂	CH ₃ CN	O ₂	76
11	1 -OTf	CH ₃ OH	O ₂	87
12	1 -Cl	CH ₃ OH	O ₂	78
13 ^[f]	1 -BPh ₄	CH ₃ OH	O ₂	98
14	1 -NO ₃	H ₂ O	O ₂	88 ^[b] (99 ^[g])

^[a]The conversion is equal to the yields of biphenyl and phenol (1:2 molar ratio) determined by ¹H NMR.

^[b]Reaction time 12 h. ^[c]No light. ^[d]Catalyst amount: 60 mol%. ^[e]Catalyst amount: 30 mol%. ^[f]**1**-BPh₄ (8 BPh₄⁻ ions per cage) and NaBPh₄ were used in 1:2 molar ratio so that the amount of the cage catalyst was 10 mol%. The yield was calculated based on the total amount of BPh₄⁻. ^[g]Reaction time 18 h.

MS, Page 17:

The anion dependence can be related to the binding affinity (Supplementary Table 1): the competing anion with strong affinity adversely influences the interactions of BPh_4^- with **1**, and the electron-donating anion like Cl^- also competes with BPh_4^- in PET. When **1-BPh₄** was used as the source of the catalytic cage, the conversion is slightly higher than that using **1-NTf₂** (entry 13), confirming a weak adverse effect of the Tf_2N^- ion.

Supplementary Information, Page S14:

1-BPh₄: To **1-BAr_F** (50 mg, 5.4 μmol , 1.0 equiv.) in 2.0 mL CH_3OH was added NaBPh_4 (15 mg, 43.2 μmol , 8.0 equiv.). 10 mL diethyl ether was added to the mixture, and a large amount of orange precipitate appeared. After filtration and washing with diethyl ether (3.0×10 mL), the precipitate was collected and dried under vacuum to obtain **1-BPh₄** (71% yield).

As expected by the reviewer, we observed the coupling products in the studies conducted in Fig. 5. We have mentioned this in the revised MS and Supplementary Information.

MS, Page 12:

The photogenerated blue solutions (Figs. 5a and 5b), either from **1-BAr_F** or from mixed **1-OTf** and BAr_4^- , showed no indication of fading if kept under N_2 overnight, but upon exposure to O_2 or air, the solutions faded rapidly concomitant with the generation of biaryls and phenols (oxidation products of BAr_4^-) (Supplementary Fig. 72). The phenomenon indicates that the blue V^{*+} radical was quenched through electron transfer (ET) to O_2 , which generated superoxide ($\text{O}_2^{\bullet-}$) for BAr_4^- oxidization (*vide infra*). The generation of $\text{O}_2^{\bullet-}$ was confirmed by spin-trapping ESR for a sample of **1-BAr_F** irradiated under O_2 . The use of 5,5-dimethyl-1-pyrroline *N*-oxide (DMPO) as spin trap led to the sharp ESR signals of the $\text{DMPO-O}_2^{\bullet-}$ adduct (Fig. 5e).

Supplementary Information, Page S54:

Supplementary Fig. 72 ^1H NMR (CD_3OD , 400 MHz, 298 K) spectra of the samples recycled from the color-faded solutions in Supplementary Fig. 71 after exposing to the air, showing the presence of aryl-coupling products in solutions.

3. There are major inconsistencies in references. A full list of authors is sometimes given, other times only first author et. al. This should be corrected.

Reply: We thank the reviewer for the careful inspection. We actually presented the referencing format following the requirement published on the *Nature Communications* website (<https://www.nature.com/ncomms/submit/how-to-submit>): “*Nature Communications* uses standard *Nature* referencing style. All authors should be included in reference lists unless there are six or more, in which case only the first author should be given, followed by 'et al.'” Nevertheless, we have rechecked all references in the revised MS to ensure the accuracy.